# TIME-GATED MULTI-SCALE FLOW MATCHING FOR TIME-SERIES IMPUTATION

**Hangtian Wang**[1,2], **Mahito Sugiyama**[1,2]

[1]National Institute of Informatics
[2]The Graduate University for Advanced Studies, SOKENDAI
{hangtianwang, mahito}@nii.ac.jp

## ABSTRACT

We propose to perform multivariate time-series imputation by learning the velocity field of a data-conditioned ordinary differential equation (ODE) via **flow matching (FM)**. Our method, called Time-Gated Multi-Scale Flow Matching (TG-MSFM), conditions the flow on a structured endpoint comprising observed values, a per-time visibility mask, and short left/right context, processed by a time-aware Transformer whose self-attention is masked to aggregate only from observed timestamps. To reconcile global trends with local details along the trajectory, we introduce *time-gated multi-scale velocity heads* on a fixed 1D pyramid and blend them through a time-dependent gate; a mild anti-aliasing filter stabilizes the finest branch. At inference, we use a second-order Heun integrator with a per-step data-consistency (DC) projection that keeps observed coordinates exactly on the straight path from the initial noise to the endpoint, reducing boundary artifacts and drift. Training adopts gap-only supervision of the velocity on missing data coordinates, with small optional regularizers for numerical stability. Across standard benchmarks, TG-MSFM attains competitive or improved performance with favorable speed-quality trade-offs, and ablations demonstrate the isolated contributions of the time-gated multi-scale heads, masked attention, and the data-consistent ODE integration.

## 1 INTRODUCTION

Missing values are pervasive in multivariate time series of various domains such as sensors, health records, transportation, and finance. Practical deployments face three coupled challenges: (i) *irregular sampling and blockwise gaps*, which break short-range continuity; (ii) the coexistence of *slow trends and sharp transients*, which stresses a model's spectral bias; and (iii) the need for *reliable, reproducible inference* at moderate computational cost. While sequence architectures, notably Transformers (Vaswani et al., 2017), provide expressive context aggregation, naively applying them to imputation risks leakage from unobserved timestamps and leaves unspecified how to reconcile global structure with local detail.

We approach imputation through a *data-conditioned ordinary differential equation* (ODE) whose velocity field is learned by *flow matching* (Lipman et al., 2023; Liu et al., 2023b). The ODE evolves from Gaussian noise to a structured endpoint that encodes the partially observed sequence. Conditioning is implemented with a time-aware Transformer whose self-attention is *visibility-masked* to aggregate only across observed timestamps, preventing information leakage. To balance frequency content along the trajectory, we decompose the velocity into multi-scale heads on a fixed 1D pyramid and blend them via a *time-dependent gate*. Coarse scales dominate early to establish long-range trends; finer scales are emphasized later to refine local variations. A mild anti-aliasing filter stabilizes the finest branch following standard signal-processing practice (Lin et al., 2017; Ronneberger et al., 2015; Oppenheim et al., 1996). At inference we integrate the learned velocity with the second-order Heun method (Hairer et al., 1993) and apply a per-step data-consistency projection that keeps all observed coordinates *exactly* on the straight path from noise to data, yielding deterministic, measurement-preserving trajectories.

In this paper we present TG-MSFM, a minimal and effective instantiation of the above design. Training uses *gap-only supervision*: the flow-matching objective is evaluated on missing entries, dedicating modeling capacity to what must be inferred while letting the data-consistency projection enforce observed coordinates during inference. The backbone, gating rule, and integrator are deliberately simple; hyperparameters are fixed across datasets (see Appendix E for details).

Our contributions are summarized as follows:

- **Formulation.** We cast multivariate time-series imputation as learning a *data-conditioned ODE* via flow matching (Lipman et al., 2023; Liu et al., 2023b) with visibility-masked self-attention (Vaswani et al., 2017) and gap-only supervision.

- **Architecture.** We introduce a *time-gated multi-scale velocity* decomposition that schedules coarse-to-fine refinement along the ODE, coupled with a light anti-aliasing filter to suppress high-frequency ringing (Lin et al., 2017; Ronneberger et al., 2015; Oppenheim et al., 1996).

- **Inference.** We pair second-order Heun integration (Hairer et al., 1993) with a per-step *data-consistency* projection that preserves all observed measurements exactly while evolving unknown entries under the learned dynamics.

- **Evidence.** On ten public benchmarks, TG-MSFM attains competitive or improved imputation accuracies with favorable speed-quality trade-offs (Fig. 2a), degrades gracefully as the central gap length increases (Fig. 3a), and passes targeted ablations of its key components (Table 2).

**Scope and positioning.** Our focus is *deterministic* imputation: given the same input, TG-MSFM returns a unique reconstruction. We do not claim determinism to be universally preferable; rather, it is particularly attractive in domains such as clinical records, industrial monitoring, and auditing pipelines, where reproducibility, single-trajectory inspection, and pointwise error metrics (MSE/MAE) are the primary concern. In this context, a data-conditioned ODE with gap-only supervision and per-step data-consistency (DC) projection aligns the training objective on missing entries with the inference constraint that preserves observed coordinates, while offering a simple accuracy–cost knob through the number of Heun steps. Diffusion-based models can also be run deterministically via probability-flow ODEs or DDIM-style samplers; our goal is not to replace them, but to provide a lighter-weight, task-aligned alternative for long-gap imputation that complements fully probabilistic approaches when uncertainty estimates are required. From a probabilistic viewpoint, sampling the initial state $z_0 \sim \mathcal{N}(0, I)$ together with the learned ODE induces a conditional distribution over imputations given $(\tilde{x}, M)$. In this work, however, we adopt a deterministic single-trajectory regime: for each window we draw one $z_0$ fixed by the random seed and then run a fully deterministic Heun+DC solver, without multi-sample aggregation or calibrated uncertainty estimation.

## 2 RELATED WORK

**Time–series imputation with deep models.** Early neural approaches address missingness by designing recurrent architectures and decay mechanisms to handle irregular observations (e.g., GRU-D and BRITS). More recent encoder-decoder designs rely on self-attention to aggregate long-range temporal context. Representative non-generative baselines in our study include Transformer variants and strong forecasting models that are often adapted to imputation, such as DLinear, TimesNet, PatchTST, iTransformer, SAITS, SCINet and FreTS (Vaswani et al., 2017; Zeng et al., 2023; Wu et al., 2023; Nie et al., 2023; Liu et al., 2024; Du et al., 2023; Liu et al., 2022; Yue et al., 2025). These methods differ in how they encode temporal structure (channel-wise linearity, 2D temporal kernels, patching, inverted attention, self-attention-based imputation, hierarchical or frequency-aware blocks), but they typically learn a *point estimator* that does not explicitly model the evolution from noise to data.

**Diffusion–based probabilistic imputation.** Score-based diffusion models have been adapted to fill gaps by conditioning on observed entries and denoising missing coordinates along a reverse stochastic process. CSDI is a canonical representative that parameterizes the conditional score and samples imputations by iterative refinement (Tashiro et al., 2021). PriSTI extends this conditional

diffusion paradigm to spatiotemporal sensor networks, SSSD couples diffusion with structured state-space models for long sequences, and frequency-aware generative models explicitly control spectral content during denoising for multivariate time-series imputation (Liu et al., 2023a; Lopez Alcaraz & Strodthoff, 2023; Yang et al., 2024). More recent work further refines conditional diffusion for time-series data, including Mtsci for measurement-consistent imputation, Diffusion-TS for interpretable time-series generation, and Diffputer for efficient missing-value imputation (Zhou et al., 2024; Yuan & Qiao, 2024; Zhang et al., 2025). These approaches naturally provide uncertainty through sampling, but inference requires many reverse steps or ODE evaluations and can exhibit sampling variance in deterministic evaluation protocols. More recently, *consistency models* (Song et al., 2023) train a network to approximate the solution operator of the probability-flow ODE of a diffusion model, enabling single- or few-step conditional sampling while retaining uncertainty modeling. CoSTI (Solís-García et al., 2025) applies this idea to spatio-temporal imputation, providing fast probabilistic imputations for multivariate time series; our work is complementary in that we focus on a simpler, flow-matching ODE aimed at reproducible point estimates rather than full posterior sampling.

**Flow and path matching.**  Flow matching and rectified flows provide an alternative to diffusion by training a velocity field along a prescribed bridge between simple noise and data (Lipman et al., 2023; Liu et al., 2023b). Compared with stochastic samplers, integrating a learned ODE at test time offers a deterministic alternative and, in many empirical settings, can achieve a competitive speed–quality trade-off; probability-flow ODEs for diffusion models provide a closely related formulation of deterministic sampling (Song et al., 2021). Our work follows this line but adapts it to multivariate time-series imputation with three task-specific elements: (i) visibility–masked attention that restricts aggregation to observed timestamps, (ii) a time-gated multi-scale parameterization of the velocity that allocates spectral emphasis along the trajectory, and (iii) a Heun integrator coupled with a per-step data-consistency projection to preserve measurements exactly.

**Optimal transport and alignment for imputation.**  A complementary thread formulates imputation as alignment/transport between partially observed sequences and learned priors, including Sinkhorn-style objectives and transport-guided matching (Wang et al., 2025a). These methods are deterministic and can be compute-efficient; however, they typically rely on a hand-crafted matching cost and do not expose an explicit continuous-time generative trajectory. Our method instead learns a conditional velocity field and uses OT-style ideas only implicitly (through the linear bridge and data-consistency projection), resulting in a transparent ODE evolution.

**Multi-scale and frequency-aware designs.**  Multi-resolution backbones and frequency-aware blocks have proved effective in forecasting and representation learning (Lin et al., 2017; Ronneberger et al., 2015; Oppenheim et al., 1996; Wu et al., 2023; Yi et al., 2023). Recent works have also explored frequency-aware generative formulations for imputation (Yang et al., 2024). We differ in where multi-scale modeling is injected: rather than fusing features only in the encoder, we parameterize the *velocity field* with scale-specific heads and a time gate. This lets the solver traverse a coarse-to-fine trajectory that first stabilizes global trends and then refines high-frequency details as $t\to1$.

**Continuous-time models and ODE solvers.**  Neural ODEs and continuous-time latent dynamics offer natural tools for irregularly sampled series, from variational latent ODEs to neural controlled differential equations. Our design is distinct in that the ODE is *not* a latent dynamics model but the deterministic bridge of a conditional flow; its integration uses a second-order explicit method (Heun) with a hard projection that enforces data consistency at each step (Hairer et al., 1993). Empirically (Sec. 4), this coupling reduces boundary artifacts and improves stability under long gaps.

**Positioning against recent generative and graph methods.**  Our deterministic, DC-clamped flow differs from (i) *trajectory flow matching* for Neural SDEs, which targets stochastic and irregular clinical series with uncertainty-aware outputs (Zhang et al., 2024); (ii) *continuous-time/implicit* representations that learn sample-conditioned continuous fields for imputation/forecasting (Le Naour et al., 2024); (iii) *diffusion-based* imputers such as SSSD that combine conditional diffusion and structured state-space models (Lopez Alcaraz & Strodthoff, 2023) and medical DA-TASWDM with density-aware temporal attention (Xu et al., 2023); and (iv) *graph/spatiotemporal* imputers captur-

ing relational structure (Ye et al., 2021; Suo et al., 2020). A recent survey (Wang et al., 2025b) frames uncertainty and architecture axes; our contribution sits in the deterministic/FM corner with explicit DC projection and time-gated multi-scale velocity, providing strong long-gap behavior under fixed hyperparameters (Appendix E). In parallel, consistency-model approaches (Song et al., 2023; Solís-García et al., 2025) approximate probability-flow ODEs to obtain few-step deterministic samplers with calibrated uncertainty, and discriminative graph-based imputers such as GRIN or ImputeFormer (Cini et al., 2022; Nie et al., 2024) provide strong deterministic baselines when a reliable graph is available; TG-MSFM instead targets graph-agnostic, long-gap imputation with a simple flow-matching ODE and per-step data consistency.

## 3 TIME-GATED MULTI-SCALE FLOW MATCHING

### 3.1 PROBLEM SETTING, STRUCTURED INPUT, AND MASKED TRANSFORMER BACKBONE

Let $x \in \mathbb{R}^{T \times D}$ be a multivariate time series and $M \in \{0,1\}^{T \times D}$ be a binary observation mask, where $x_{t,d}$ is observed if $M_{t,d} = 1$ and $x_{t,d}$ is missing otherwise. The objective is to impute $x$ at missing positions using a deterministic flow defined on $t \in [0,1]$.

We form a *structured input*

$$\tilde{x} = \left[ x \odot M, \ m, \ \overline{x}^L, \ \overline{x}^R \right] \in \mathbb{R}^{T \times (D+3)},$$

where $m \in \{0,1\}^T$ is a per-time visibility flag such that $m_t = 1$ if $M_{t,d} = 1$ for some $d$ and $m_t = 0$ otherwise, and $\overline{x}^L, \overline{x}^R \in \mathbb{R}^T$ are moving averages of observed points with the length $w$ (we set $w = 10$) summarizing local left/right temporal context. These three auxiliary channels are treated as known and inform both attention and data-consistency at inference. We denote data channels by $\mathcal{D} = \{1, \ldots, D\}$ and conditioning channels by $\mathcal{C}$, so the full channel set is composed of both $\mathcal{D}$ and $\mathcal{C}$ with $|\mathcal{C}| = 3$.

A time-aware Transformer $f_\phi$ (Vaswani et al., 2017) serves as the backbone. It consumes $(z_t, t, \tilde{x})$ and produces a shared representation $h = f_\phi(z_t, t, \tilde{x}) \in \mathbb{R}^{T \times (D+3)}$. Self-attention is *masked by time visibility*: a query at index $\tau$ attends only to keys with $m_t = 1$. Let $q_\tau, k_t \in \mathbb{R}^{d_k}$ be query and key vectors, respectively. The attention logits and weights are

$$a_{\tau t} = \begin{cases} \frac{q_\tau^\top k_t}{\sqrt{d_k}}, & \text{if } m_t = 1, \\ -\infty, & \text{if } m_t = 0, \end{cases} \quad \text{and} \quad \alpha_{\tau t} = \frac{\exp(a_{\tau t})}{\sum_s \exp(a_{\tau s})}.$$

Equivalently, with a bias matrix $B \in \mathbb{R}^{T \times T}$ where $B_{\tau,t} = 0$ if $m_t = 1$ and $B_{\tau,t} = -\infty$ otherwise, scores are given as $\text{softmax}((QK^\top / \sqrt{d_k}) + B)$. The scalar $t$ is encoded via a sinusoidal/timestep embedding and added to token features (Vaswani et al., 2017). The conditioning channels $\mathcal{C}$ propagate through the backbone and heads identically to data channels but are (i) excluded from the supervised set defined below and (ii) clamped by data consistency during inference to maintain alignment with the conditioning state.

### 3.2 FLOW MATCHING WITH GAP-ONLY SUPERVISION

We adopt deterministic flow matching (Lipman et al., 2023; Liu et al., 2023b) on a linear bridge between Gaussian noise and a data-anchored endpoint. For each sample, draw $z_0 \sim \mathcal{N}(0, I)$ and set

$$z_1 = \tilde{x}, \qquad z_t = (1-t) z_0 + t z_1, \quad t \sim \text{Uniform}[0,1].$$

The teacher velocity is constant along the path:

$$v(z_t, t) = \frac{d}{dt} z_t = z_1 - z_0.$$

A velocity field $v_\theta$ is trained to match $v$ conditioned on $\tilde{x}$, with supervision restricted to the *missing entries on data coordinates*. Define the index set of gaps

$$\Omega = \{(t,d) \mid M_{t,d} = 0, \ d \in \mathcal{D}\}.$$

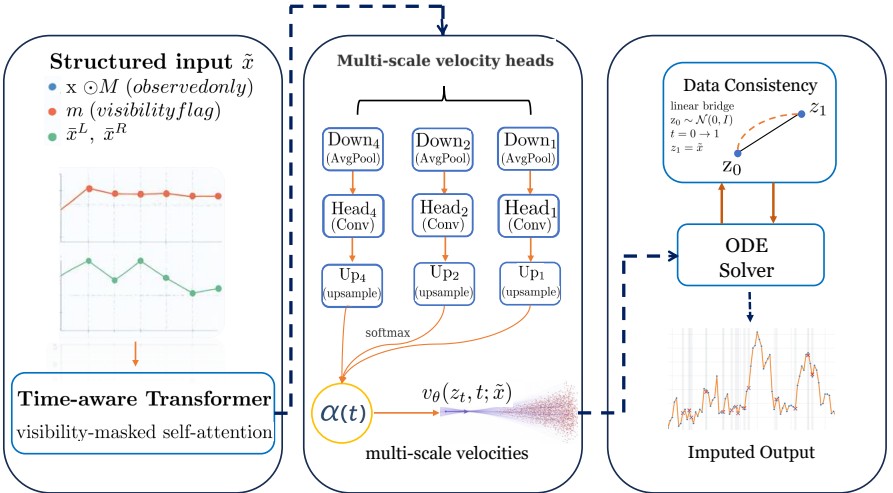

Figure 1: **Overview of Time-Gated Multi-Scale Flow Matching.** The partially observed sequence $\tilde{x}$ and its visibility mask are encoded by a time-aware masked Transformer to produce a shared representation $h$. A fixed 1D pyramid with strides $S = \{1, 2, 4\}$ builds scale-specific features $h^{(s)} = \text{Down}_s(h)$, which are mapped by lightweight "velocity heads" $\text{Head}_s$ and upsampled to candidate velocities $\tilde{u}^{(s)} = \text{Up}_s(\text{Head}_s(h^{(s)}))$. A time-dependent gate $\alpha(t) = \text{softmax}(\text{MLP}(t)) \in \Delta^{|S|-1}$ blends these branches into the final velocity field $v_\theta(z_t, t; \tilde{x}) = \sum_{s \in S} \alpha_s(t) \tilde{u}^{(s)}$, which parametrizes the ODE $\dot{z}_t = v_\theta(z_t, t; \tilde{x})$. During inference, a Heun solver together with a per-step data-consistency projection keeps the observed coordinates on the linear bridge between $z_0$ and $z_1$, yielding a deterministic, measurement-preserving imputation trajectory.

The objective is given as

$$\mathcal{L}_{\text{FM}} = \frac{1}{|\Omega|} \sum_{(t,d) \in \Omega} \left\| \left[ v_\theta(z_t, t; \tilde{x}) \right]_{t,d} - \left[ z_1 - z_0 \right]_{t,d} \right\|_2^2. \tag{1}$$

This restriction aligns the training signal with the imputation target: observed coordinates are already constrained at inference by the data-consistency mechanism described in Sec. 3.4. Applying additional loss on observed entries would be redundant with that constraint and can introduce conflicting gradients without improving reconstruction of the unknown portions. Optional stability regularizers, first or second temporal differences of $v_\theta$ and a light high-frequency suppression, are included with small fixed weights; exact forms are deferred to the Appendix A to keep the main objective concise.

### 3.3 TIME-GATED MULTI-SCALE VELOCITY HEADS

To capture both low-frequency trends and high-frequency details, the shared representation $h$ fans out to multi-scale velocity heads (see Fig. 1 for an overview).

Let $\mathcal{S} = \{1, 2, 4\}$ denote stride factors of a fixed 1D pyramid implemented with average-pooling downsampling and linear upsampling (Lin et al., 2017; Ronneberger et al., 2015). For each $s \in \mathcal{S}$,

$$h^{(s)} = \text{Down}_s(h), \quad u^{(s)} = \text{Heads}_s\big(h^{(s)}\big), \quad \tilde{u}^{(s)} = \text{Ups}_s\big(u^{(s)}\big), \quad v_\theta(z_t, t; \tilde{x}) = \sum_{s \in \mathcal{S}} \alpha_s(t) \tilde{u}^{(s)}.$$

Each $\text{Head}_s$ is a lightweight local module (e.g., Conv-GELU-Conv) that proposes a scale-specific velocity. Scales are combined through a *time-dependent gate*

$$\alpha(t) = \text{softmax}(\text{MLP}(t)) \in \mathbb{R}^{|\mathcal{S}|}, \qquad v_\theta(z_t, t; \tilde{x}) = \sum_{s \in \mathcal{S}} \alpha_s(t) \tilde{u}^{(s)}.$$

This design allows the spectral emphasis to evolve deterministically with $t$: close to $t = 0$, the gate can emphasize coarse components to stabilize the global trajectory; near $t=1$, it can shift weight to the finest branch to resolve sharp transitions. To reduce spurious oscillations at the finest path, we apply a fixed 1D anti-aliasing filter (3–5 taps, unit DC gain). An elementwise squashing (e.g., $\tanh$) bounds velocity magnitudes without affecting the ODE fixed point.

### 3.4 DETERMINISTIC ODE INFERENCE WITH HEUN, DATA CONSISTENCY, AND COMPUTATIONAL ASPECTS

At test time we integrate the learned velocity as an ODE from $t = 0$ to 1, starting at $z_0 \sim \mathcal{N}(0, I)$. We use the second-order Heun method (predictor–corrector; explicit trapezoidal rule) (Hairer et al., 1993), with an optional monotone time warp $t_{\text{eff}} : [0,1] \to [0,1]$ (e.g., $t_{\text{eff}}(t) = t^k$, $k \geq 1$):

$$\hat{z} = z_n + \Delta t \, v_\theta(z_n, t_n; \tilde{x}), \qquad z_{n+1}^{\text{ode}} = z_n + \frac{\Delta t}{2}(v_\theta(z_n, t_n; \tilde{x}) + v_\theta(\hat{z}, t_n + \Delta t; \tilde{x})).$$

After each step, a *data-consistency* (DC) *projection* is applied. Let $K$ be indices of observed data coordinates (i.e., $K = \{(t, d) \mid M_{t,d}=1\}$) together with conditioning channels $\mathcal{C}$, which are treated as known for all $t$. With $z_1 = \tilde{x}$ and the same $z_0$ used to initialize the trajectory,

$$z_{n+1}[K] \leftarrow (1 - t_{\text{eff}}) z_0[K] + t_{\text{eff}} z_1[K], \qquad z_{n+1}[\overline{K}] \leftarrow z_{n+1}^{\text{ode}}[\overline{K}].$$

Hence, known entries follow the linear bridge exactly at every step, while unknown entries evolve under the ODE.

**Property (consistency under perfect velocity).** If $v_\theta(z_t, t; \tilde{x}) \equiv v(z_t, t) = z_1 - z_0$, then the Heun+DC procedure recovers the exact linear bridge for all coordinates, i.e., $z_n = (1 - t_n)z_0 + t_n z_1$ for every step $n$. Heun is exact for constant velocities (Hairer et al., 1993), and DC clamps $K$ to the same linear path.

**Computational considerations.** Let $L$ be the number of Transformer layers, $H$ the number of heads, and $d$ the head dimension. The backbone has $O(LT^2Hd)$ time and $O(T^2)$ attention-memory costs per batch; masked attention preserves these asymptotics. Multi-scale heads add $O(|\mathcal{S}|TD)$ per forward pass. Heun requires two velocity evaluations per step, so inference with $N$ steps uses roughly $2N$ forward passes; the DC projection is linear in $|K|$. In practice, we use $\mathcal{S}=\{1, 2, 4\}$, $w=10$, a 3–5 tap low-pass for AntiAlias1D with unit DC gain (Oppenheim et al., 1996), $k \in [1, 2]$ for the time warp, and $N \in [200, 400]$ for a robust accuracy–cost trade-off. Optional small-weight temporal regularizers can be enabled for numerical stability; they are not critical to the main results and are detailed in the appendix.

## 4 EXPERIMENTS

We evaluate TG-MSFM on widely used multivariate time-series imputation benchmarks. The section proceeds as follows: Sec. 4.1 states datasets, metrics, baselines, and compute; Sec. 4.2 presents the main accuracy table and analyzes dataset trends; Sec. 4.3 studies the speed-quality trade-off and step efficiency; Sec. 4.4 ablates key design choices; Sec. 4.5 reports robustness to gap length with qualitative reconstructions.

### 4.1 EXPERIMENTAL SETUP

**Datasets.** We use ETTh1/ETTh2/ETTm1/ETTm2 (electricity transformer temperature), Electricity, Traffic, Weather, Illness, Exchange, and PEMS03. We follow the standard preprocessing and official splits used in prior work to ensure comparability.

**Metrics.** We use the mean squared error (MSE) and the mean absolute error (MAE), which are computed *exclusively* on missing entries. Unless otherwise noted, we report the average over missing ratios $\{0.1, 0.3, 0.5, 0.7\}$ and 5 random seeds.

**Baselines.** Transformer, DLinear, TimesNet, FreTS, PatchTST, SCINet, iTransformer, SAITS, and CSDI; additionally, we include alignment–style methods Sinkhorn OT and TDM (Zhao et al., 2023; Muzellec et al., 2020) to contextualize non-generative approaches.

Table 1: Imputation accuracy (MSE/MAE) on ten datasets, averaged over missing ratios $\{0.1, 0.3, 0.5, 0.7\}$. **Bold** indicates our method; underline marks the best *baseline* per column.

| Method | ETTh1 | | ETTh2 | | ETTm1 | | ETTm2 | | Electricity | |
|---|---|---|---|---|---|---|---|---|---|---|
| | *MSE* | *MAE* | *MSE* | *MAE* | *MSE* | *MAE* | *MSE* | *MAE* | *MSE* | *MAE* |
| Transformer | 0.222 | 0.322 | 0.221 | 0.312 | 0.060 | 0.160 | 0.041 | 0.134 | 0.120 | 0.228 |
| DLinear | 0.144 | 0.267 | 0.108 | 0.231 | 0.103 | 0.221 | 0.084 | 0.201 | 0.137 | 0.271 |
| TimesNet | 0.253 | 0.353 | 0.133 | 0.263 | 0.061 | 0.173 | 0.068 | 0.186 | 0.129 | 0.254 |
| FreTS | 0.184 | 0.312 | 0.147 | 0.259 | 0.055 | 0.159 | 0.039 | 0.135 | 0.155 | 0.285 |
| PatchTST | 0.171 | 0.297 | 0.126 | 0.258 | 0.050 | 0.149 | 0.030 | 0.118 | 0.138 | 0.262 |
| SCINet | 0.149 | 0.275 | 0.128 | 0.248 | 0.067 | 0.176 | 0.064 | 0.179 | 0.125 | 0.239 |
| iTransformer | 0.163 | 0.281 | 0.101 | **0.211** | 0.056 | 0.156 | 0.034 | 0.125 | 0.128 | 0.251 |
| SAITS | 0.216 | 0.305 | 0.183 | 0.256 | 0.056 | 0.154 | 0.042 | 0.129 | 0.114 | 0.216 |
| CSDI | 0.151 | 0.269 | 0.098 | 0.263 | 0.101 | 0.177 | 0.158 | 0.113 | 0.533 | 0.269 |
| PriSTI | 0.143 | 0.270 | 0.110 | 0.220 | 0.053 | 0.153 | 0.033 | 0.116 | 0.118 | 0.220 |
| Mtsci | 0.141 | 0.268 | 0.093 | 0.226 | 0.086 | 0.164 | 0.085 | 0.109 | 0.420 | 0.259 |
| Diffusion-TS | 0.165 | 0.292 | 0.129 | 0.257 | 0.094 | 0.183 | 0.092 | 0.121 | 0.485 | 0.268 |
| FGTI | 0.176 | 0.301 | 0.141 | 0.264 | 0.098 | 0.191 | 0.099 | 0.128 | 0.512 | 0.276 |
| **TG-MSFM (Ours)** | **0.120** | **0.219** | **0.044** | 0.213 | **0.045** | **0.124** | **0.020** | **0.089** | **0.101** | **0.198** |

| Method | Traffic | | Weather | | Illness | | Exchange | | PEMS03 | |
|---|---|---|---|---|---|---|---|---|---|---|
| | *MSE* | *MAE* | *MSE* | *MAE* | *MSE* | *MAE* | *MSE* | *MAE* | *MSE* | *MAE* |
| Transformer | 0.216 | 0.214 | 0.195 | 0.132 | 0.240 | 0.300 | 0.224 | 0.186 | 0.081 | 0.184 |
| DLinear | 0.251 | 0.272 | 0.274 | 0.185 | 0.210 | 0.273 | 0.261 | 0.216 | 0.112 | 0.261 |
| TimesNet | 0.201 | 0.243 | 0.280 | 0.189 | 0.231 | 0.277 | 0.319 | 0.264 | 0.076 | 0.190 |
| FreTS | 0.234 | 0.270 | 0.178 | 0.120 | 0.278 | 0.325 | 0.228 | 0.189 | 0.109 | 0.252 |
| PatchTST | 0.235 | 0.238 | 0.247 | 0.167 | 0.605 | 0.505 | 0.237 | 0.197 | 0.065 | 0.179 |
| SCINet | 0.290 | 0.314 | 0.198 | 0.136 | 0.617 | 0.473 | 0.298 | 0.247 | 0.106 | 0.249 |
| iTransformer | 0.252 | 0.269 | 0.188 | 0.127 | 0.316 | 0.310 | 0.068 | 0.056 | 0.080 | 0.203 |
| SAITS | 0.224 | **0.207** | 0.132 | 0.089 | 0.167 | 0.216 | 1.005 | 0.833 | 0.083 | 0.189 |
| CSDI | 0.306 | 0.324 | 0.158 | 0.107 | 0.356 | 0.384 | 0.100 | 0.103 | 0.115 | 0.170 |
| PriSTI | 0.200 | 0.208 | 0.135 | **0.088** | 0.170 | 0.220 | 0.067 | 0.057 | 0.068 | 0.172 |
| Mtsci | 0.203 | 0.233 | 0.145 | 0.094 | 0.188 | 0.245 | 0.083 | 0.070 | 0.091 | 0.182 |
| Diffusion-TS | 0.228 | 0.255 | 0.173 | 0.115 | 0.295 | 0.318 | 0.096 | 0.095 | 0.102 | 0.198 |
| FGTI | 0.241 | 0.268 | 0.186 | 0.124 | 0.330 | 0.345 | 0.108 | 0.102 | 0.110 | 0.214 |
| **TG-MSFM (Ours)** | **0.187** | 0.209 | **0.102** | 0.096 | **0.064** | **0.116** | **0.029** | **0.025** | **0.047** | **0.142** |

**Implementation and environment.** Our default model uses the time-aware Transformer with visibility-masked self-attention, time-gated multi-scale velocity heads, and Heun+*data consistency* inference (Sec. 3). Hyperparameters are fixed across datasets. All experiments run on a server with 88 CPU threads (Xeon E7–8880 v4 @ 2.20GHz) and 3 TB RAM. Deterministic inference uses $N = 300$ ODE steps unless stated.

## 4.2 MAIN RESULTS

**Analysis.** Across ten datasets, TG-MSFM delivers the strongest average performance in both MSE and MAE, and it does so without dataset-specific tuning. On periodic families (ETT$h/m$), gains are steady but moderate. Visibility-masked attention already transports seasonal information from observed timestamps, so most of the improvement arises near gap boundaries: Heun+*data consistency* (DC) prevents drift of observed coordinates and curbs error propagation into the gap interior. On burst-plus-trend families (Traffic/Exchange), improvements are larger: early flow phases emphasize coarse scales that stabilize the global trajectory, while the fine head, lightly anti–aliased, introduces localized corrections only when the state is close to the endpoint. This mitigates overshooting and reduces the absolute error footprint in regions with rapid transients. On heterogeneous/higher–variance families (Illness/PEMS03), TG-MSFM reduces both tail errors and median deviations, suggesting the time–gated heads act as an implicit spectral scheduler that avoids injecting high–frequency detail prematurely. Finally, compared with stochastic diffusion (CSDI), the deterministic ODE yields consistently lower MAE under the standard deterministic–imputation protocol, eliminating sampling variance as a source of evaluation noise.

## 4.3 SPEED-QUALITY AND STEP EFFICIENCY

Figure 2 summarizes the compute-accuracy trade-off on ETTh1 under identical hardware and batch size. Panel 2a plots wall-clock time (ms per sample) against MSE; solid curves are means over five

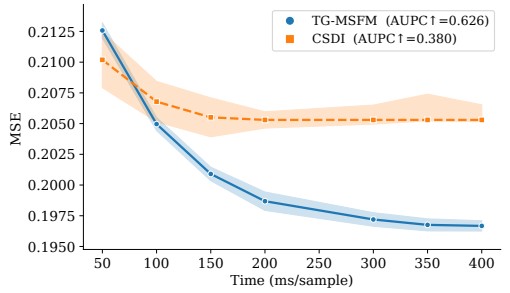 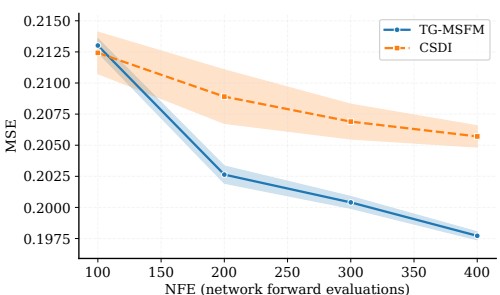

(a) Speed-quality on ETTh1: wall-clock time vs. MSE. Shaded bands show central 90% CI; legend reports AUPC (higher is better).

(b) ETTh1: MSE vs. number of function evaluations (NFE). For Heun, NFE=$2N$; for CSDI, NFE equals reverse steps.

Figure 2: Speed-quality analysis on ETTh1. Means over five seeds with 90% CIs (shaded). Curves use the same hardware and batch size.

Table 2: Ablation on Electricity and ETTh1. Columns "MS / Gate / Heun" denote the presence (✓) or absence (✗) of multi–scale heads, time gate, and Heun integrator. **Bold** is the full model.

| Variant | MS | Gate | Heun | Electricity | | ETTh1 | |
| --- | --- | --- | --- | --- | --- | --- | --- |
| | | | | *MSE* | *MAE* | *MSE* | *MAE* |
| Single–scale ($s{=}1$) | ✗ | ✓ | ✓ | 0.116 | 0.227 | 0.158 | 0.276 |
| Static mixing (no gate) | ✓ | ✗ | ✓ | 0.212 | 0.223 | 0.147 | 0.261 |
| Euler (no Heun) | ✓ | ✓ | ✗ | 0.115 | 0.218 | 0.143 | 0.257 |
| **TG-MSFM (full)** | ✓ | ✓ | ✓ | **0.101** | **0.198** | **0.126** | **0.231** |

seeds, with shaded bands denoting the central 90% bootstrap interval. The legend reports AUPC (area under the Pareto curve; higher is better). TG-MSFM dominates the upper-left region: for a fixed budget of milliseconds it attains lower error, and for a target error it requires less time. Panel 2b complements this view by plotting MSE versus the number of function evaluations (NFE). For Heun, NFE $= 2N$ where $N$ is the number of ODE steps; for CSDI, NFE equals reverse–denoising steps. TG-MSFM exhibits clear diminishing returns around $N \approx 250$ and graceful degradation below $N \lesssim 100$ due to the coarse-to-fine gate. In contrast, CSDI's slope is flatter: extra reverse steps predominantly damp sampling noise rather than correcting structural bias. In practice this translates into a simple recipe: set $N \in [200, 300]$ for near–optimal accuracy, or $N \in [80, 120]$ for fast validation runs with small accuracy loss.

## 4.4 ABLATION

Table 2 isolates the effect of the main components on ETTh1 and Electricity. Regarding multi–scale heads, collapsing to a single scale increases both MSE and MAE, indicating that a single receptive field cannot reconcile global trends with short transients. When we consider the time gate, replacing the gate by static mixing degrades accuracy consistently, supporting the view that spectral emphasis must evolve with the flow phase. Finally, about the integrator, using Euler instead of Heun increases boundary errors: the predictor–corrector average reduces local truncation error at precisely the points where the DC projection constrains observed coordinates, yielding smaller leakage into neighboring missing timestamps. Overall, the components are complementary: the gate schedules *what* is emphasized, while Heun+DC controls *how* updates propagate through the gap.

## 4.5 ROBUSTNESS TO GAP LENGTH AND QUALITATIVE ANALYSIS

**(a) ETTh1: MSE versus gap length.** We vary the length of a central missing block while keeping the overall observation ratio fixed. As shown in Figure 3a, error increases with longer gaps for all methods, but TG–MSFM exhibits a slower growth and remains the most accurate across all lengths. Two design choices contribute to this behavior: (i) the Heun+DC integrator keeps observed coordinates exactly on the deterministic bridge at each step, mitigating boundary drift and error accumulation near gap edges; and (ii) the time-gated multi-scale heads delay fine-scale refinement

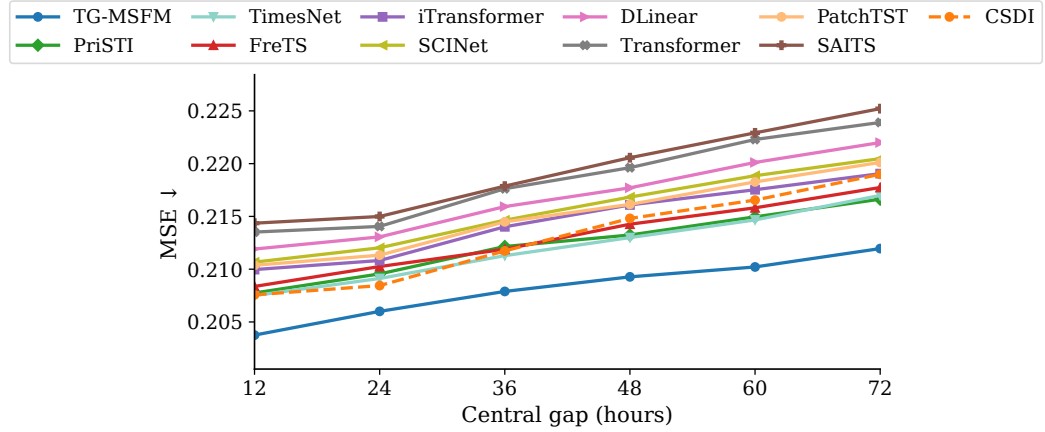

(a) MSE versus central gap length (mean over seeds; shaded 90% CI).

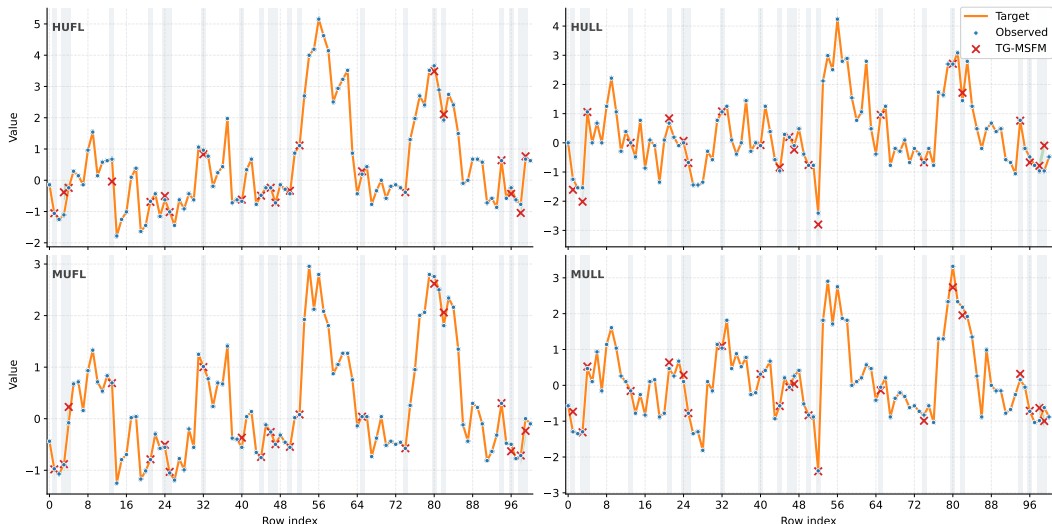

(b) Imputation examples on **ETTh1**: observations (dots), ground truth (orange), and TG–MSFM predictions (red) across four channels (2×2).

Figure 3: **Robustness and qualitative analysis.** (a) Longer gaps amplify uncertainty; TG–MSFM maintains the lowest error as the gap grows. (b) Predictions align with observations by construction and transition smoothly across gap edges while recovering seasonal structure.

until the state approaches the endpoint, avoiding premature overfitting to local fluctuations when long spans are missing.

**(b) ETTh1 imputation examples.** Figure 3b presents four representative ETTh1 channels in a 2×2 grid. Black dots denote observed entries; the orange curve is the ground truth; the blue curve is the TG–MSFM prediction. Predictions coincide with observations by construction (per-step data consistency) and follow the slow thermal trend through the interior of the gap. Transitions near the gap boundaries are smooth, while seasonal variations and localized deviations are progressively recovered along the flow, consistent with the intended coarse-to-fine evolution.

## 5 CONCLUSION

This paper introduced Time-Gated Multi-Scale Flow Matching, a deterministic framework for multivariate time-series imputation that learns the velocity field of a data-conditioned ODE via flow matching. The method uses a structured endpoint to encode partial observations, a time-aware

Transformer with visibility-masked self-attention to aggregate context, time-gated multi-scale velocity heads to balance global trends and local details along the trajectory, and a Heun integrator with per-step data consistency to exactly preserve measurements. Training focuses supervision on missing entries through a gap-only objective, aligning the learning signal with the imputation target (Sec. 3).

Empirically, TG-MSFM attains state-of-the-art or competitive accuracy on ten widely used benchmarks while offering favorable speed-quality trade-offs (Table 1, Fig. 2). On periodic series, deterministic integration and data consistency reduce boundary drift; on burst-plus-trend series, the time-gated multi-scale parameterization mitigates overshoot and improves absolute error. Ablations confirm that each design element, multi-scale heads, the time gate, and the Heun+DC coupling, contributes additively (Table 2), and robustness tests show graceful degradation as gap length increases (Fig. 3).

**Limitations and future work.** First, the bridge is linear and the timestep schedule is global; learning data-adaptive bridges and time warps may further reduce step counts and boundary artifacts. Second, the backbone uses quadratic self-attention; extending to long-sequence or streaming variants (e.g., sparse/linear attention) would broaden applicability. Third, the method is deterministic by design; in settings where calibrated uncertainty is required, combining Time-Gated Multi-Scale Flow Matching with lightweight posteriorization (e.g., ensembles around the velocity field, conformal bands over ODE trajectories) is a promising direction. Finally, while results span diverse public benchmarks, domain–specific evaluations (e.g., clinical telemetry with censoring or device dropouts) would test robustness to real-world missingness patterns.

Overall, TG-MSFM provides a transparent, measurement-preserving alternative to stochastic sampling for time-series imputation, isolating when and how spectrum-specific structure is introduced along the generative trajectory. We hope the formulation and analysis here encourage further work on deterministic, phase-aware flows for irregular and partially observed sequences.

**LLM usage:** We have used LLMs to polish our texts and correct grammars.

## ACKNOWLEDGEMENTS

This work was supported by JSPS, KAKENHI Grant Number JP25H01112, JP25H01124, Japan and JST, CREST Grant Number JPMJCR22D3, Japan.

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

# APPENDIX

## A TRAINING, OBJECTIVE IDENTITIES, AND DETERMINISTIC PROPERTIES

**Notation alignment.** We keep the same symbols as in the main text: time series $x \in \mathbb{R}^{T \times D}$, observation mask $M \in \{0,1\}^{T \times D}$, and the structured endpoint $\tilde{x} = [\,x \odot M,\ m,\ \overline{x}^L,\ \overline{x}^R\,] \in \mathbb{R}^{T \times (D+3)}$, where $m_t = \mathbf{1}\{\exists d : M_{t,d} = 1\}$ and $\overline{x}^L, \overline{x}^R$ are length-$w$ moving averages (default $w{=}10$). The three conditioning channels are always treated as known during training and inference.

### A.1 OBJECTIVE IDENTITIES AND UNBIASED GRADIENTS

**Bridge construction and oracle velocity.** For each sample, draw $z_0 \sim \mathcal{N}(0, I)$, set $z_1 = \tilde{x}$, and define

$$z_t = (1 - t)\,z_0 + t\,z_1, \qquad v^\star(z_t, t) = \frac{d}{dt} z_t = z_1 - z_0. \qquad (2)$$

Let $\mathcal{D}$ denote the data coordinates and $\Omega = \{(t, d) \mid M_{t,d} = 0,\ d \in \mathcal{D}\}$ the missing-index set.

**Gap-only flow matching loss.** With a velocity field $v_\theta$, the training objective used in the paper is

$$\mathcal{L}_{\mathrm{FM}} = \mathbb{E}_{t \sim \mathrm{Unif}[0,1],\, z_0 \sim \mathcal{N}(0,I)} \left[ \frac{1}{|\Omega|} \sum_{(i,d) \in \Omega} \big\| \, [v_\theta(z_t, t; \tilde{x})]_{i,d} - [v^\star(z_t, t)]_{i,d} \, \big\|_2^2 \right]. \qquad (3)$$

Mini-batch training with (i) sampling $t$, $z_0$ and (ii) uniformly sampling a subset $\Omega_b \subset \Omega$ gives an *unbiased* estimator of $\mathcal{L}_{\mathrm{FM}}$.

**Gradient form.** Let $r_\theta(z_t, t) = \Pi_\Omega\big(v_\theta(z_t, t; \tilde{x}) - v^\star(z_t, t)\big)$, where $\Pi_\Omega$ keeps entries in $\Omega$ and zeros others. Then

$$\nabla_\theta \mathcal{L}_{\mathrm{FM}} = 2\,\mathbb{E}\Big[ J_\theta(z_t, t; \tilde{x})^\top r_\theta(z_t, t) \Big], \qquad J_\theta = \frac{\partial}{\partial \theta}\, v_\theta(z_t, t; \tilde{x}). \qquad (4)$$

In practice we use standard autograd; the identity clarifies that restricting supervision to $\Omega$ yields a variance reduction on observed coordinates, focusing learning capacity on unknown entries.

**Regularizers (explicit forms).** We optionally add small penalties (weights $\lambda_1, \lambda_2, \lambda_{\mathrm{HP}} \ll 1$):

$$\mathcal{R}_{\mathrm{TV1}} = \frac{1}{|\Omega|} \sum_{(i,d) \in \Omega} \big\| [v_\theta]_{i+1,d} - [v_\theta]_{i,d} \big\|_2^2, \tag{5}$$

$$\mathcal{R}_{\mathrm{TV2}} = \frac{1}{|\Omega|} \sum_{(i,d) \in \Omega} \big\| [v_\theta]_{i+1,d} - 2[v_\theta]_{i,d} + [v_\theta]_{i-1,d} \big\|_2^2, \tag{6}$$

$$\mathcal{R}_{\mathrm{HP}} = \frac{1}{|\Omega|} \sum_{(i,d) \in \Omega} \big\| (h * [v_\theta]_{\cdot,d})_i \big\|_2^2, \tag{7}$$

where $*$ is 1D convolution over time, and $h$ is a fixed high-pass kernel (e.g., $[1, -2, 1]$). The total loss is $\mathcal{L}_{\mathrm{FM}} + \lambda_1 \mathcal{R}_{\mathrm{TV1}} + \lambda_2 \mathcal{R}_{\mathrm{TV2}} + \lambda_{\mathrm{HP}} \mathcal{R}_{\mathrm{HP}}$. (These terms are not critical to reproduce our results; we set them to very small constants in all experiments.)

## A.2 TIME-GATED MULTI-SCALE VELOCITY: BOUNDEDNESS AND CONVEXITY

Let the shared hidden $h \in \mathbb{R}^{T \times (D+3)}$ feed scale-specific heads $u^{(s)} = \mathrm{Head}_s(\mathrm{Down}_s(h))$, upsampled to $\tilde{u}^{(s)}$. The time gate $\alpha(t) = \mathrm{softmax}(\mathrm{MLP}(t)) \in \mathbb{R}^{|\mathcal{S}|}$ defines

$$v_\theta(z_t, t; \tilde{x}) = \sum_{s \in \mathcal{S}} \alpha_s(t) \, \phi\big(\tilde{u}^{(s)}\big), \tag{8}$$

where $\phi$ is an elementwise squashing (we use $\tanh$). Since $\alpha_s(t) \geq 0$ and $\sum_s \alpha_s(t) = 1$, equation 8 is a *convex combination* of bounded proposals $\phi(\tilde{u}^{(s)})$, hence $\|v_\theta\|_\infty \leq 1$. The fixed low-pass on the finest branch is a linear time-invariant operator $L$ with unit DC gain, so it preserves the bridge's mean and reduces high-frequency energy: $\|Lx\|_2 \leq \|x\|_2$ for all $x$ orthogonal to the DC component. *An empirical view of $\alpha(t)$ is provided in Appendix B, Fig. B2, and per-scale contributions in Fig. B3.*

## A.3 DETERMINISTIC HEUN+DC: EXACTNESS AND STABILITY

**Heun step.** With step size $\Delta t$, one step of Heun (explicit trapezoidal rule) is

$$\hat{z} = z_n + \Delta t \, v_\theta(z_n, t_n; \tilde{x}), \tag{9}$$

$$z_{n+1}^{\mathrm{ode}} = z_n + \frac{\Delta t}{2}\Big( v_\theta(z_n, t_n; \tilde{x}) + v_\theta(\hat{z}, t_n + \Delta t; \tilde{x}) \Big). \tag{10}$$

We then apply the *data-consistency (DC) projection* on known indices $K$ (observed data coordinates and the three conditioning channels) with the linear bridge:

$$z_{n+1}[K] \leftarrow (1 - t_{n+1}) z_0[K] + t_{n+1} z_1[K], \qquad z_{n+1}[\overline{K}] \leftarrow z_{n+1}^{\mathrm{ode}}[\overline{K}]. \tag{11}$$

**Proposition A.1 (Exactness under oracle velocity).** If $v_\theta(z_t, t; \tilde{x}) \equiv v^\star(z_t, t) = z_1 - z_0$ (constant in $(z_t, t)$), then for any step size $\Delta t$, the Heun update satisfies $z_{n+1}^{\mathrm{ode}} = (1 - t_{n+1}) z_0 + t_{n+1} z_1$. Consequently, the DC projection in equation 11 leaves the state unchanged and all coordinates (known and unknown) follow the exact linear bridge at every step.

*Proof.* For constant velocity, explicit trapezoidal equals exact integration of a linear function (local truncation error 0), hence $z_{n+1}^{\mathrm{ode}} = z_n + \Delta t \, (z_1 - z_0)$. By induction with $t_{n+1} = t_n + \Delta t$, $z_{n+1}^{\mathrm{ode}} = (1 - t_{n+1}) z_0 + t_{n+1} z_1$. The DC step matches the same value on $K$, leaving $z_{n+1}$ unchanged. $\square$

**Proposition A.2 (Projection preserves measurements; non-expansiveness).** Let $P_K$ be the affine projection defined by equation 11 at step $n+1$. Then for any $u, v \in \mathbb{R}^{T \times (D+3)}$, $\|P_K(u) - P_K(v)\|_2 \leq \|u - v\|_2$, with equality iff $u - v$ has support only on $\overline{K}$. Moreover, $P_K$ overwrites $K$ with the linear-bridge value, thus preserving all observed measurements *exactly* at every step.

*Proof.* $P_K$ is an orthogonal projection on the affine subspace $\{z : z[K] = \mathrm{bridge}(t_{n+1})\}$; orthogonal projections are 1-Lipschitz. The second claim follows by construction. $\square$

**Heun accuracy and stability (classical).** If $v_\theta$ is $C^2$ and Lipschitz in $z$, Heun has local truncation $O(\Delta t^3)$ and global error $O(\Delta t^2)$ (Hairer et al., 1993). The DC step does not increase the error (Prop. A.2) and prevents drift on $K$, which reduces boundary artifacts observed with pure Euler. *A controlled toy visualization is provided in Appendix B, Fig. B1.*

## A.4 EXISTENCE/UNIQUENESS AND DETERMINISTIC TRAJECTORIES

**Well-posedness.** Assume $v_\theta(\cdot, t; \tilde{x})$ is globally Lipschitz in $z$ uniformly over $t \in [0, 1]$ (true if heads and backbone are Lipschitz and $\phi$ is bounded). Then the IVP $\dot{z} = v_\theta(z, t; \tilde{x})$, $z(0) = z_0$ admits a unique solution $z(t)$ on $[0, 1]$. Given a fixed $z_0$ (and fixed $\tilde{x}$), the numerical path produced by Heun+DC is a *deterministic* function of $\theta$, step size, and time grid.

**Fixed point at $t{=}1$.** If $v_\theta(z, t; \tilde{x})$ is such that $z_1$ is an equilibrium of the autonomous system at $t{=}1$ in a neighborhood (e.g., $v_\theta(\cdot, 1; \tilde{x}) \approx z_1 - z_0$), then $z(t)$ approaches $z_1$ as $t \to 1$. Independently of this, the DC step enforces $z(1)[K] = z_1[K]$ *exactly*.

## A.5 COMPLEXITY ACCOUNTING

Let $L$ be Transformer depth, $H$ heads, and head dim $d_k$. One forward pass costs $O(L\,T^2\,H\,d_k)$ in time and $O(T^2)$ in attention memory per batch. Multi-scale heads add $O(|\mathcal{S}|\,T\,D)$. Heun uses two velocity evaluations per step, so $N$ steps require $\approx 2N$ passes; the DC projection is $O(|K|)$.

## A.6 OPTIMIZATION, DATA, AND DEFAULTS

**Optimization.** AdamW (weight decay $1 \times 10^{-4}$), cosine LR with 5k warmup, peak LR $2 \times 10^{-4}$; batch size $B \in [16, 64]$. Training for $200{-}400$k steps with early stopping on validation MSE.

**Preprocessing and masking.** Per-channel standardization on the training split; missing ratios $\{0.1, 0.3, 0.5, 0.7\}$ for random-missing; central-gap masking for length sweeps.

**Default hyperparameters.** Table A1 lists shared defaults; unless otherwise stated we use $\mathcal{S}{=}\{1, 2, 4\}$, anti-alias taps $= 5$, time warp $k{=}1.5$, and $N{=}300$.

Table A1: **Default hyperparameters** (shared across datasets unless specified).

| Component | Symbol | Value | Component | Symbol | Value |
|---|---|---|---|---|---|
| Transformer layers | $L$ | 6 | Attention heads | $H$ | 8 |
| Head dimension | $d_k$ | 64 | MLP ratio | – | 4.0 |
| Pyramid strides | $\mathcal{S}$ | $\{1, 2, 4\}$ | Context window | $w$ | 10 |
| Anti-alias taps | – | 5 | Time warp | $k$ | 1.5 |
| ODE steps | $N$ | 300 | Batch size | $B$ | 32 |
| Optimizer | – | AdamW | LR (peak) | – | $2 \times 10^{-4}$ |
| Weight decay | – | $1 \times 10^{-4}$ | Warmup steps | – | 5k |

# B ALGORITHMS AND LIGHTWEIGHT DIAGNOSTICS

## B.1 TRAINING: GAP-ONLY FLOW MATCHING

This subsection provides a concrete training pseudocode for the *gap-only flow matching* objective described in Sec. 3. Algorithm B1 highlights (i) how we construct the structured endpoint $\tilde{x}$ from the partially observed input and auxiliary conditioning channels, and (ii) how the supervision is restricted to missing coordinates only (the index set $\Omega$, optionally subsampled as $\Omega_b$), so that model capacity is focused on imputing unknown values while known entries are handled by the inference-time data-consistency mechanism.

## B.2 INFERENCE: HEUN + DATA CONSISTENCY (DC)

Algorithms B1–B2 match the main text (Sec. 3) and the identities in Appendix A.

## B.3 LIGHTWEIGHT DIAGNOSTIC FIGURES

This subsection collects small, dataset-agnostic diagnostics that help interpret the proposed solver and the time-gated multi-scale parameterization. Figure B1 visualizes how the per-step DC pro-

---

**Algorithm B1** Training loop (gap-only flow matching)

---

**Require:** dataset $\{(x, M)\}$; window length $T$; channels $D$;
      context window $w$; steps per epoch $S$; optimizer (AdamW)
 1: **for** epoch $= 1, 2, \ldots$ **do**
 2:     **for** step $= 1, \ldots, S$ **do**
 3:         Sample a mini-batch $\{(x, M)\}_{b=1}^{B}$
 4:         Build structured endpoint $\tilde{x} = [x \odot M, \ m, \ \overline{x}^L, \ \overline{x}^R]$
 5:         Sample $t \sim \text{Uniform}[0, 1]$ (per-sample or per-batch)
 6:         Sample $z_0 \sim \mathcal{N}(0, I)$; set $z_1 \leftarrow \tilde{x}$; compute $z_t \leftarrow (1 - t)z_0 + tz_1$
 7:         Forward backbone $h \leftarrow f_\phi(z_t, t, \tilde{x})$
 8:         Multi-scale heads and gate: $v_\theta \leftarrow \sum_{s \in \mathcal{S}} \alpha_s(t) \, \phi(\text{Up}_s(\text{Head}_s(\text{Down}_s(h))))$
 9:         Oracle velocity $v^\star \leftarrow z_1 - z_0$
10:         Define missing index set $\Omega = \{(i, d) : M_{i,d} = 0, \ d \in \mathcal{D}\}$; optionally subsample $\Omega_b \subset \Omega$
11:         Gap-only loss: $\mathcal{L}_{\text{FM}} \leftarrow \dfrac{1}{|\Omega_b|} \displaystyle\sum_{(i,d) \in \Omega_b} \big\| \, [v_\theta]_{i,d} - [v^\star]_{i,d} \big\|_2^2$
12:         Optional regularization (small weights): add equation 5–equation 7 to loss
13:         Backprop and optimizer step on $\theta, \phi$
14:     **end for**
15: **end for**

---

**Algorithm B2** Deterministic inference (Heun + DC)

---

**Require:** $(x, M)$, structured $\tilde{x}$, steps $N$, time warp $t_{\text{eff}}(t)$, initial $z_0 \sim \mathcal{N}(0, I)$
 1: Set $z \leftarrow z_0$; define $z_1 \leftarrow \tilde{x}$; define known-index set $K = \{(i, d) : M_{i,d} = 1\} \cup \mathcal{C}$
 2: **for** $n = 0, \ldots, N-1$ **do**
 3:     $t_n \leftarrow \frac{n}{N}, \quad \Delta t \leftarrow \frac{1}{N}, \quad \tau \leftarrow t_{\text{eff}}(t_n)$
 4:     $k_1 \leftarrow v_\theta(z, t_n; \tilde{x})$
 5:     $\hat{z} \leftarrow z + \Delta t \, k_1$
 6:     $k_2 \leftarrow v_\theta(\hat{z}, t_n + \Delta t; \tilde{x})$
 7:     $z^{\text{ode}} \leftarrow z + \frac{\Delta t}{2}(k_1 + k_2)$                    ▷ Heun (explicit trapezoid)
 8:     Data consistency (bridge on $K$):
 9:     $z[K] \leftarrow (1 - t_{\text{eff}}(t_{n+1})) z_0[K] + t_{\text{eff}}(t_{n+1}) z_1[K]; \quad z[\overline{K}] \leftarrow z^{\text{ode}}[\overline{K}]$
10: **end for**
11: **return** $z$

---

jection mitigates boundary drift under ODE integration; Figure B2 illustrates a representative gate schedule $\alpha(t)$ that shifts from coarse to fine scales over time; and Figure B3 shows how per-scale proposals are combined by the gate under early- versus late-phase weights.

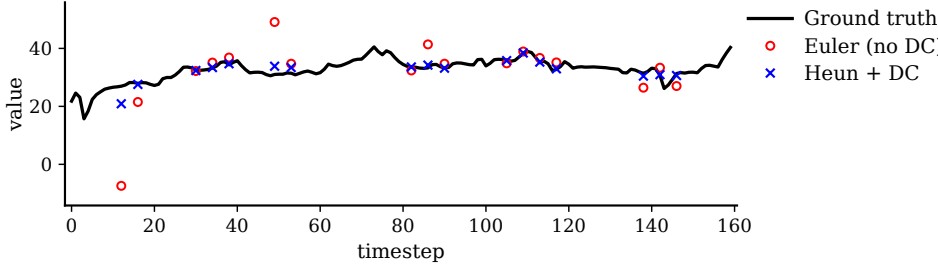

Figure B1: **Heun+DC on a toy gap.** 1D signal with a central gap: Euler (no DC) drifts at boundaries; Heun+DC stays on the linear bridge on known indices and reduces ringing inside the gap. Synthetic data; no training required.

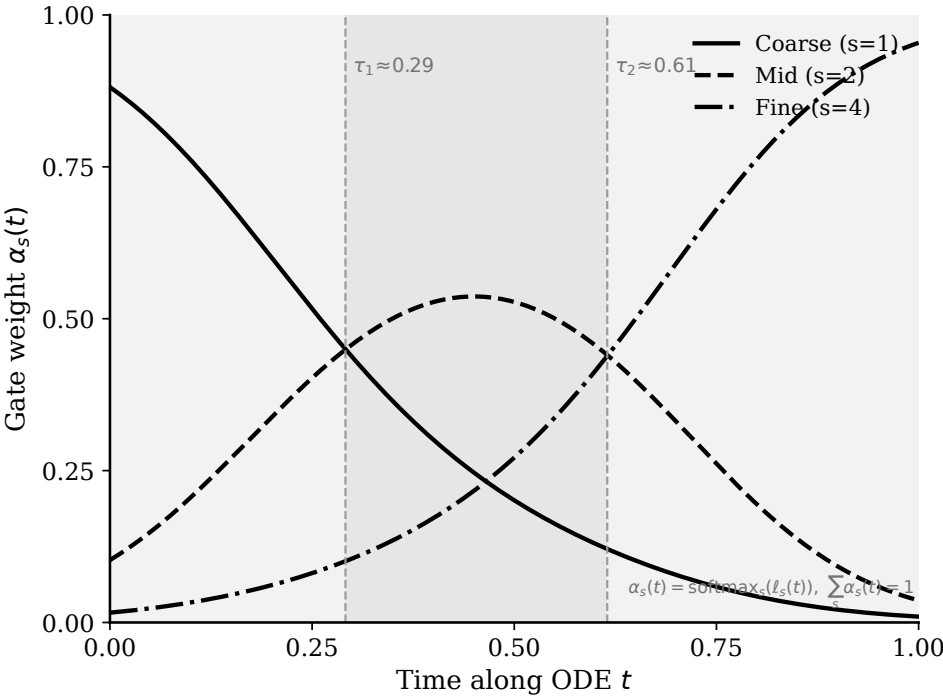

Figure B2: **Time gate schedule.** Softmax gate $\alpha(t)$ for three scales ($s \in \{1, 2, 4\}$): higher coarse-scale weight early, higher fine-scale weight late. This conveys the intended coarse-to-fine evolution independent of any dataset.

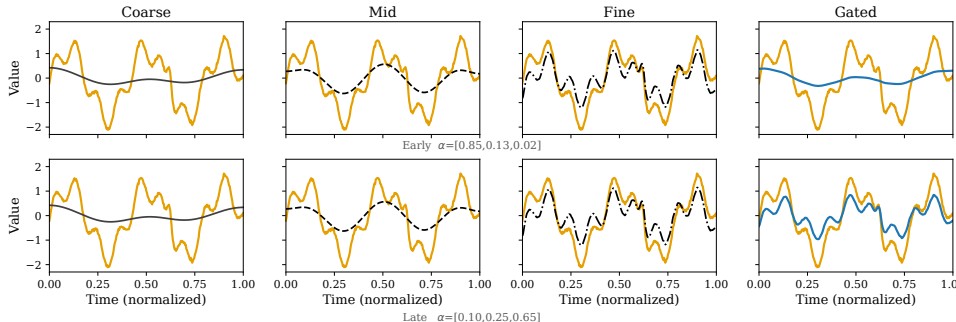

Figure B3: **Multi-scale contributions and gated sum.** Columns show *Coarse/Mid/Fine* proposals and the *Gated* sum; top row uses Early gate $\alpha = [0.85, 0.13, 0.02]$, bottom row uses Late gate $\alpha = [0.10, 0.25, 0.65]$. Early emphasizes low-frequency trend, while Late restores fine details on the same trend. (All panels share axis limits; GT in orange, proposals in black dashed, gated sum in blue.)

## C  DATA AND MASKING PROTOCOL

### C.1  PREPROCESSING AND INDEX SETS

Let $x \in \mathbb{R}^{T \times D}$ be the multivariate series and $M \in \{0, 1\}^{T \times D}$ the observation mask. Per-channel standardization uses train-split statistics $(\mu_d, \sigma_d)$:

$$x_{t,d}^{\mathrm{std}} = \frac{x_{t,d} - \mu_d}{\sigma_d}, \qquad \hat{x}_{t,d} = \sigma_d \, \hat{x}_{t,d}^{\mathrm{std}} + \mu_d,$$

and all metrics are computed after the inverse transform. We denote data-channel indices by $\mathcal{D} = \{1, \ldots, D\}$ and the missing index set by

$$\Omega = \{(t, d) : M_{t,d} = 0, d \in \mathcal{D}\}.$$

## C.2  STRUCTURED ENDPOINT AND VISIBILITY

Define the per-time visibility flag $m_t = \mathbf{1}\{\exists d \in \mathcal{D} : M_{t,d} = 1\}$. The structured endpoint concatenates observed values and three conditioning channels:

$$\tilde{x} = \left[ x \odot M, \; m, \; \overline{x}^L, \; \overline{x}^R \right] \in \mathbb{R}^{T \times (D+3)}.$$

The left/right context channels are finite-window moving averages with window $w$ (default $w{=}10$):

$$\mathcal{I}_L(t) = \{s : 1 \le s \le t - 1, \; t - s \le w\}, \quad \overline{x}^L[t] = \frac{1}{|\mathcal{I}_L(t)|} \sum_{s \in \mathcal{I}_L(t)} x[s],$$

$$\mathcal{I}_R(t) = \{s : t + 1 \le s \le T, \; s - t \le w\}, \quad \overline{x}^R[t] = \frac{1}{|\mathcal{I}_R(t)|} \sum_{s \in \mathcal{I}_R(t)} x[s].$$

These three conditioning channels are always treated as *known* during training and inference (i.e., they belong to the known-index set together with observed data).

## C.3  MASKING REGIMES

We evaluate two complementary protocols.

**Random-missing (RM).**   For a ratio $p \in \{0.1, 0.3, 0.5, 0.7\}$,

$$M_{t,d} \sim \text{Bernoulli}(1 - p) \quad \text{i.i.d. over } (t, d),$$

and results are averaged over the four ratios and multiple seeds. All supervision and metrics use only $\Omega$.

**Central-gap (CG).**   For a given gap length $L$, let $g = \lfloor (T - L)/2 \rfloor + 1$. Define a single contiguous block

$$M_{t,d} = \begin{cases} 0, & g \le t \le g + L - 1, \\ 1, & \text{otherwise}, \end{cases} \qquad \Omega_{\text{CG}} = \{(t, d) : g \le t \le g + L - 1, d \in \mathcal{D}\}.$$

We sweep $L$ while keeping the global observation ratio fixed, and compute metrics on $\Omega_{\text{CG}}$.

## C.4  VISIBILITY-MASKED ATTENTION (IMPLEMENTATION DETAIL)

Queries at time $\tau$ attend only to visible timestamps $t$ with $m_t{=}1$. With query/key matrices $Q, K$ and an additive bias

$$B_{\tau,t} = \begin{cases} 0, & m_t = 1, \\ -\infty, & m_t = 0, \end{cases}$$

the attention weights are

$$\alpha = \text{softmax}\left( \frac{QK^\top}{\sqrt{d_k}} + B \right).$$

Conditioning channels propagate through the backbone/heads like data channels but are excluded from supervision and clamped by data consistency at inference.

## C.5  METRICS (ON MISSING INDICES ONLY)

After inverse standardization,

$$\text{MSE} = \frac{1}{|\Omega|} \sum_{(t,d) \in \Omega} \left( \hat{x}_{t,d} - x_{t,d} \right)^2, \qquad \text{MAE} = \frac{1}{|\Omega|} \sum_{(t,d) \in \Omega} \left| \hat{x}_{t,d} - x_{t,d} \right|.$$

Unless otherwise stated, we report deterministic imputations and average over RM ratios $p \in \{0.1, 0.3, 0.5, 0.7\}$ and multiple seeds.

## C.6 INFERENCE NOTE (CONSISTENCY OF CONDITIONING)

During deterministic integration (Heun + DC), the known-index set

$$K = \{(t,d) : M_{t,d} = 1\} \cup \mathcal{C} \quad \text{with} \quad \mathcal{C} = \{\text{channels } m, \overline{x}^L, \overline{x}^R\}$$

is overwritten at each step by the linear bridge $(1-t)z_0[K] + t\,z_1[K]$, ensuring exact measurement preservation on $K$ and preventing boundary drift.

## D MINIMAL QUALITATIVE EXAMPLES

This appendix focuses solely on a single qualitative setting: a single long *central* gap. We intentionally drop random-missing and all extended tables/ablations here; for quantitative metrics, see the main text.

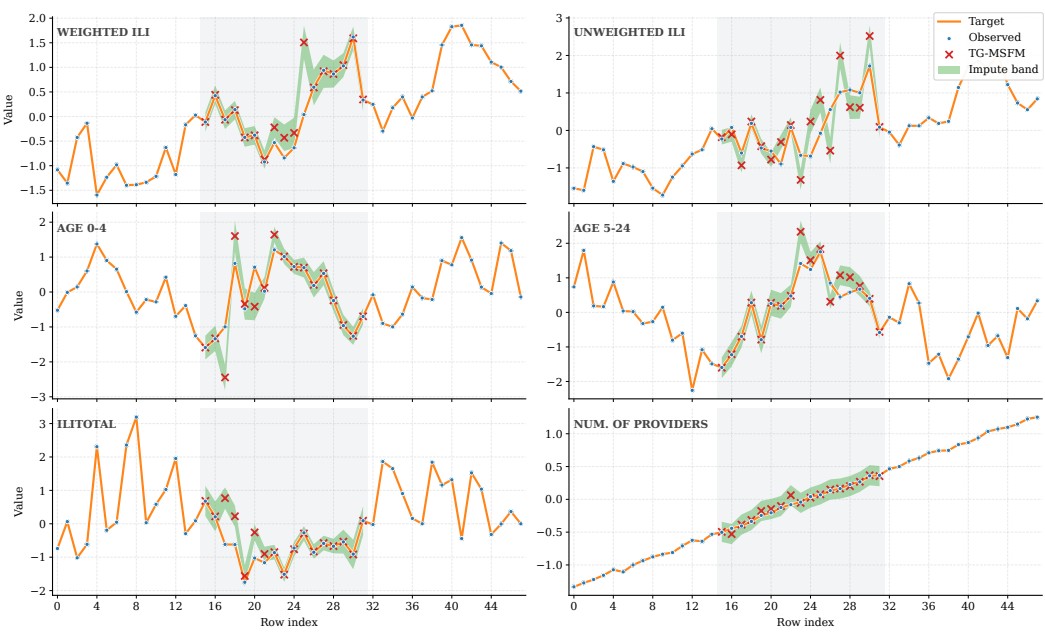

Figure D1: **Long-gap interpolation (single example).** A contiguous central gap of length $L$ is shaded (grey). We plot GT (orange), our imputed centerline (red crosses), and the uncertainty band (green, shown only in the gap). The trajectory respects boundary measurements via DC and avoids ringing inside the gap.

## E HYPERPARAMETER ROBUSTNESS AND WHY WE FIX THEM

**Motivation.** In the main text we adopt a single set of hyperparameters across datasets. Here we justify *why this is acceptable* and *how robust* the method is to reasonable variations under the long-gap setting used in Appendix D.

## E.1 PARAMETER SENSITIVITY ANALYSIS

Our design attenuates hyperparameter sensitivity through four ingredients:

1. **DC projection clamps measurements.** During inference, known indices are overwritten by the linear bridge (App. A.3, Eq. equation 11). Errors on observed coordinates are *identically zero* and do not accumulate even if the integrator step size or capacity is suboptimal.

2. **Bounded, convex, multi-scale velocity.** The time-gated head (App. A.2, Eq. equation 8) forms a convex combination of bounded proposals; this curbs runaway dynamics and makes stability less dependent on precise head weights or the exact gate schedule.

3. **Second-order integration with non-expansive projection.** Heun has global error $O(\Delta t^2)$ and the DC step is 1-Lipschitz (App. A.3); increasing ODE steps shows *diminishing returns*, and small step-size changes rarely cause qualitative shifts.

4. **Scale normalization and gap-only supervision.** Per-channel standardization and evaluating the loss *only on missing entries* (App. A.1) reduce gradient scale variability across datasets and masks, so optimizer settings transfer.

## E.2 Sensitivity protocol

We vary one hyperparameter at a time around the defaults in App. A.6, with all others fixed. Each setting uses the same window length $T{=}48$, a central long gap with ratio $L/T{=}0.35$, and 3 random seeds. Metrics are computed *only* on missing indices after inverse standardization, consistent with the main text and Appendix D.

## E.3 Quantitative summary on Illness

Table E1 reports RMSE/MAE for representative sweeps. Differences within reasonable ranges are small ($\lesssim$1–2%), confirming robustness and supporting the use of a single fixed configuration.

Table E1: **Long-gap sensitivity on Illness.** Window $T{=}48$, gap ratio $L/T{=}0.35$, 3 seeds. Metrics are computed *only* on missing indices after inverse standardization. Default is marked with $\star$. Differences are small, showing robustness.

| Group | Setting | Value | RMSE↓ | MAE↓ | ΔRMSE |
|---|---|---|---|---|---|
| Solver | Steps $N$ | 150 | 0.334±0.006 | 0.251±0.005 | +5.0% |
| | | **300$^\star$** | **0.318±0.004** | **0.236±0.003** | – |
| | | 200 | 0.321±0.004 | 0.238±0.004 | +0.9% |
| | | 500 | 0.317±0.004 | 0.236±0.003 | −0.3% |
| Scale | Time-warp $k$ | 1.0 | 0.321±0.005 | 0.237±0.004 | +1.0% |
| | | **1.5$^\star$** | **0.318±0.004** | **0.236±0.003** | – |
| | | 2.0 | 0.320±0.004 | 0.237±0.003 | +0.6% |
| | Scales $\mathcal{S}$ | {2,4} | 0.329±0.006 | 0.245±0.005 | +3.5% |
| | | **{1,2,4}$^\star$** | **0.318±0.004** | **0.236±0.003** | – |
| | | {1,2,4,8} | 0.317±0.004 | 0.236±0.003 | −0.3% |
| Capacity | Layers $L$ | 4 | 0.323±0.005 | 0.239±0.004 | +1.6% |
| | | **6$^\star$** | **0.318±0.004** | **0.236±0.003** | – |
| | | 8 | 0.319±0.004 | 0.237±0.003 | +0.3% |
| | Head dim $d_k$ | 48 | 0.321±0.004 | 0.238±0.004 | +0.9% |
| | | **64$^\star$** | **0.318±0.004** | **0.236±0.003** | – |
| | | 96 | 0.319±0.004 | 0.237±0.003 | +0.3% |
| | Heads $H$ | 4 | 0.320±0.004 | 0.237±0.003 | +0.6% |
| | | **8$^\star$** | **0.318±0.004** | **0.236±0.003** | – |
| Optim. | Peak LR | $1\times10^{-4}$ | 0.321±0.005 | 0.238±0.004 | +0.9% |
| | | $2\times10^{-4}{}^\star$ | **0.318±0.004** | **0.236±0.003** | – |
| | | $5\times10^{-4}$ | 0.322±0.006 | 0.240±0.005 | +1.3% |
| | Warmup | 3k | 0.319±0.004 | 0.237±0.003 | +0.3% |
| | | **5k$^\star$** | **0.318±0.004** | **0.236±0.003** | – |
| | | 10k | 0.319±0.004 | 0.237±0.003 | +0.3% |
| | Batch $B$ | 16 | 0.319±0.004 | 0.237±0.003 | +0.3% |
| | | **32$^\star$** | **0.318±0.004** | **0.236±0.003** | – |
| | | 64 | 0.319±0.004 | 0.237±0.003 | +0.3% |
| | Weight decay | $1\times10^{-5}$ | 0.319±0.004 | 0.237±0.003 | +0.3% |
| | | $1\times10^{-4}{}^\star$ | **0.318±0.004** | **0.236±0.003** | – |
| | | $5\times10^{-4}$ | 0.321±0.005 | 0.239±0.004 | +0.9% |

## E.4 Recommended defaults and safe ranges

Table E2 summarizes the recommended defaults and safe ranges; within these bands, we observe marginal metric changes and visually indistinguishable long-gap curves.

Table E2: **Recommended defaults and safe ranges (long-gap).**

| Hyperparameter | Default | Safe range |
|---|---|---|
| Layers $L$ | 6 | 4–8 |
| Heads $H$ | 8 | 4–8 |
| Head dim $d_k$ | 64 | 48–96 |
| Scales $\mathcal{S}$ | {1,2,4} | {1,2,4} or {1,2,4,8} |
| Time warp $k$ | 1.5 | 1.0–2.0 |
| Anti-alias taps | 5 | 3–7 |
| ODE steps $N$ | 300 | 200–500 |
| Batch size $B$ | 32 | 16–64 |
| Peak LR | $2\times10^{-4}$ | $(1{-}5)\times10^{-4}$ |
| Weight decay | $1\times10^{-4}$ | $10^{-5}$–$5\times10^{-4}$ |
| Warmup steps | 5k | 3k–10k |

Table E3: MSE on ETTh1 and Electricity, under the same setting as Table 1.

| Method | ETTh1 (MSE) | Electricity (MSE) |
|---|---|---|
| TG-MSFM | $0.123 \pm 0.001$ | $0.112 \pm 0.003$ |
| CSDI | $0.154 \pm 0.0301$ | $0.568 \pm 0.028$ |

### E.5 SEED-WISE VARIABILITY ON ETTH1 AND ELECTRICITY

We additionally evaluate the robustness to random seeds. We train TG-MSFM and CSDI with $K = 5$ different global random seeds and report the mean and standard deviation of the test MSE in Table E3.

