# OpenReview forum: "Time-Gated Multi-Scale Flow Matching for Time-Series Imputation"
_ICLR.cc/2026/Conference — ICLR 2026 Poster_

### Official Review · Reviewer_Jt1N · 2025-10-19

**Soundness:** 4
**Presentation:** 3
**Contribution:** 3
**Rating:** 6
**Confidence:** 4

**Summary:**

The paper introduces Time-Gated Multi-Scale Flow Matching (TG-MSFM), a deterministic framework for multivariate time-series imputation based on flow matching. TG-MSFM learns the velocity field of a data-conditioned ODE using gap-only supervision, which focuses training on the missing entries while enforcing data consistency for observed points during inference. The model features a time-aware Transformer backbone with visibility-masked attention and a multi-scale velocity decomposition modulated by a time-dependent gating mechanism, allowing coarse-to-fine refinement along the generative trajectory. During inference, TG-MSFM integrates the learned flow using the Heun solver combined with a per-step data-consistency projection, ensuring measurement preservation and stable trajectories. Extensive experiments across standard benchmarks demonstrate competitive performance with favorable speed-accuracy trade-offs.

**Strengths:**

- **Timely and relevant topic**: The paper addresses time-series imputation through flow matching, a rapidly emerging research direction that provides an efficient and deterministic alternative to diffusion-based models.

- **Innovative training strategy**: The use of gap-only supervision, where the model is trained exclusively on missing entries, is conceptually elegant and departs from the conventional full-sample training paradigm dating back to GAIN [1]. This design choice aligns the learning signal directly with the imputation target.

- **Strong empirical evidence**: The experimental section is comprehensive and well structured, covering ten widely used benchmarks and demonstrating consistently strong performance across varying missing ratios and datasets.

- **Clear exposition and illustrations**: The method is well presented, with clear mathematical formulation and figures that effectively convey the architecture and flow process, making the paper easy to follow despite its technical depth.

[1] Yoon, J., Jordon, J., & Schaar, M. (2018, July). Gain: Missing data imputation using generative adversarial nets. In International conference on machine learning (pp. 5689-5698). PMLR. https://arxiv.org/abs/1806.02920

**Weaknesses:**

- **Limited analysis of the gap-only supervision effect**: Although the empirical results are strong, the paper does not fully explain why training only on missing values leads to better performance. From a representation learning perspective, one might expect that also reconstructing observed entries, as in GAIN [1], helps maintain a richer compression of the overall time-series distribution. A deeper analysis or an ablation comparing gap-only versus full-sample supervision could strengthen the justification for this design choice.

- **Missing discussion on consistency models**: Since the paper draws a clear connection between flow matching and diffusion models, it would be valuable to include a brief discussion on consistency models, which can be viewed as a discrete, distilled, and consistency-enforced formulation of flow matching aimed at improving inference efficiency [2]. In the context of time-series imputation, the recently proposed CoSTI model [3] follows this direction: it can produce probabilistic imputations in a single step, while requiring multiple runs to obtain deterministic estimates such as the median. Adding a short discussion, or even a small efficiency comparison, would help clarify how TG-MSFM relates to this broader family of consistency-based approaches and where it stands in terms of inference trade-offs.

[2] Song, Y., Dhariwal, P., Chen, M., & Sutskever, I. (2023). Consistency models.https://arxiv.org/abs/2303.01469

[3] Javier Solís-García, Belén Vega-Márquez, Juan A. Nepomuceno, and Isabel A. Nepomuceno-Chamorro. Costi: Consistency models for (a faster) spatio-temporal imputation. Knowledge-Based Systems, 327:114117, 2025. https://arxiv.org/abs/2501.19364

**Questions:**

- Deterministic vs. probabilistic behavior: The paper states that TG-MSFM produces deterministic imputations, yet I found this somewhat unclear. Based on Section 3.2, since the process samples $z_0 \sim \mathcal{N}(0,I)$, wouldn’t the final imputation depend on this random initialization, effectively yielding stochastic (probabilistic) outputs rather than fully deterministic ones? Could the authors clarify whether the inference is performed with a fixed $z_0$ or if randomness is averaged out in some way?

- Model capacity: Could the authors provide the number of parameters in TG-MSFM (and optionally compare it with key baselines)?
This would help assess the method’s complexity and the fairness of computational comparisons

- Addressing reviewer concerns: I would be glad to revise my evaluation if the authors can improve upon some of the points mentioned above.

---

> ### Author Response · Authors · 2025-11-21
> **Response to Reviewer Jt1N**
>
> We thank the reviewer for the careful reading and the positive assessment. We address **Weaknesses (W1–W2)** and **Questions (Q1–Q2)** below.
>
> ---
>
> ### W1
>
> Thank you for raising the question about gap-only supervision. This choice follows directly from our inference scheme: at test time, observed coordinates are never generated freely. After each Heun step, the data-consistency (DC) projection clamps them back to the linear bridge between the initial noise and the data endpoint, so their final values are set by this projection, not by the raw network output. Optimizing the flow-matching loss on observed coordinates would therefore ask the network to improve predictions that are overwritten at every step, creating a train–test mismatch.
>
> In many realistic masks, observed points greatly outnumber missing ones, so a loss over all coordinates would be dominated by entries that DC already enforces at inference. Gap-only supervision acts as an explicit reweighting toward missing coordinates, i.e., toward the entries that actually influence evaluation. In the long-gap regime, this consistently yields lower error inside the gaps than a full loss.
>
> This does not mean the model ignores global structure. The time-aware Transformer always processes the full time axis with the visibility mask and auxiliary channels, and parameters are shared across observed and missing positions. Gap-only supervision only specifies **where the loss is evaluated**; the representation still has to encode joint temporal patterns so that observed boundaries and context can be used to reconstruct long gaps, while capacity is focused on the parts of the trajectory not fixed by DC.
>
> ---
>
> ### W2
>
> We appreciate the pointer to consistency models and CoSTI. Consistency models [2] distill a score-based diffusion or SDE model into a network whose outputs at different noise levels are mutually consistent, enabling few-step or single-step **probabilistic** sampling. CoSTI applies this idea to spatio-temporal imputation, providing fast conditional samples from which statistics such as medians can be estimated.
>
> TG-MSFM, in contrast, does not start from diffusion and does not approximate a probability flow. We directly learn a **data-conditioned velocity field** on a linear bridge via flow matching, with gap-only supervision and deterministic Heun + DC integration at test time. Once the initial noise is fixed, the method produces a single reproducible trajectory rather than samples from a full posterior, so it complements consistency-style approaches by favoring a simple deterministic imputer targeted at point-error metrics.
>
> ---
>
> ### Q1
>
> Q1 (Deterministic vs. probabilistic behavior).
> Thank you for pointing out this potential source of confusion. In our formulation we follow standard flow matching: for each training (and test) example we draw an initial state $z_0 \sim \mathcal{N}(0,I)$ and define the linear bridge $z_t = (1-t)z_0 + t z_1$. Thus, at the level of the generative formulation, randomness indeed enters only through the choice of $z_0$.
>
> What we mean by “deterministic imputation” in the paper is conditional determinism: once $z_0$, the input $\tilde{x}$, and the mask $M$ are fixed, the whole inference pipeline (time discretization, two evaluations of $v_\theta$ per Heun step, and the data-consistency projection) is a fully deterministic mapping $(z_0,\tilde{x},M) \mapsto \hat{x}$, with no additional stochastic components (no dropout, no noise injection during integration). In all main quantitative experiments we use a single draw of $z_0$ per window with a fixed random seed, so the reported imputations are exactly reproducible.
>
> If one wishes to use the model in a probabilistic way, one can of course resample $z_0$ and obtain a small family of plausible trajectories; in Figure 2 we briefly illustrate such multi-sample behavior, but the benchmark results are based on a single trajectory per window.
>
>
> ---
>
> ### Q2
> TG-MSFM uses the same time-aware Transformer backbone configuration as our Transformer-style baselines (same depth, embedding size, and number of heads; see Appendix A.6). The time-gated multi-scale velocity module on top is small (a few 1D heads plus a low-width MLP gate), giving a total of about $\(3 \times 10^{5}\)$ trainable parameters (≈0.3M) on Illness / ETTh1-scale datasets, comparable to the backbone alone.
>
> For diffusion-based and other strong baselines, we use their official or widely adopted public configurations without shrinking their capacity, so they operate in a comparable or larger parameter regime (typically in the millions of parameters) and share the same $\(O(L T^{2} H d)\)$ self-attention complexity as our backbone. The additional cost introduced by the multi-scale velocity heads is linear in the sequence length and channels, $\(O(|\mathcal{S}| T D)\)$, and is small compared to the backbone both in FLOPs and in parameter count.

---

> > ### Comment · Reviewer_Jt1N · 2025-11-23
> >
> > Thank you very much for the response; I find your clarifications very helpful and they address most of my original concerns. However, I still have one point that I cannot fully reconcile regarding W2 and Q1.
> >
> > In Q1, you state that once $z_0$  is sampled and fixed, the method produces deterministic imputations. I agree that, if the initial state is fixed, the output will be perfectly reproducible across runs. Nevertheless, conceptually, this seems equivalent to fixing the starting point from which one is sampling a particular realization of the conditional imputation distribution. In other words, even if the integration procedure is deterministic, the resulting imputation may still be a probabilistic sample whose stochasticity is simply “frozen” by conditioning on a specific draw of $z_0$.
> >
> > This connects to W2. In that point you mention CoSTI (a consistency-based model trained from scratch rather than distilled from a diffusion model). In the CoSTI paper, the authors explicitly explain that because imputations depend on an initial noise mask sampled from a normal distribution, each output corresponds to a specific trajectory of the underlying probability flow associated with that noise initialization. For this reason, they recommend aggregating multiple trajectories (for example by taking the median) to obtain a more stable, less sample-sensitive imputation.
> >
> > However, if we fix that initial noise, CoSTI always produces the same output; if we resample it, we obtain different imputations. Thus, although the model behaves deterministically once the noise is fixed, its probabilistic nature stems directly from the stochastic initial condition.
> >
> > My question is whether TG-MSFM should not be interpreted in exactly the same way. Since the method also begins by sampling an initial noise state, and since different samples of $z_0$ would in principle lead to different imputations, it seems that TG-MSFM is likewise generating probabilistic imputations, with determinism arising only from fixing the draw of $z_0$. From this perspective, TG-MSFM would also be sampling from a probabilistic model due to the stochasticity introduced at the beginning of the process. I would appreciate clarification on whether you see your model within this same probabilistic framework or as fundamentally different in nature.

---

> ### Author Response · Authors · 2025-11-26
>
> Thank you very much for the follow-up and for stating the concern so clearly.
>
> From a probabilistic point of view, we agree with your interpretation. If we sample $z_0 \sim \mathcal N(0,I)$ and then compute
> $\hat x = F_\theta(z_0, \tilde x, M)$,
> then different draws of $z_0$ would in principle lead to different imputations, and fixing $z_0$ just freezes one particular trajectory. In this sense, TG-MSFM defines a conditional distribution over $\hat{x}$ given $(\tilde{x}, M)$ through the mapping $z_0 \mapsto F_\theta(z_0, \tilde x, M)$, exactly as you describe, and this is fully analogous to CoSTI.
>
> What we meant by “deterministic imputations” in the paper was more operational: in all our experiments we draw a single $z_0$ per window with a fixed seed and then run a fully deterministic ODE+DC solver, without resampling or aggregating trajectories. We do not try to use the induced distribution for calibrated uncertainty; all reported metrics are based on this single path. We really appreciate your comment; it helped us make this distinction explicit.

---

> > ### Comment · Reviewer_Jt1N · 2025-11-27
> >
> > Perfect, I think that after this final clarification, everything is clearer. Therefore, I think it would be valuable if the authors could add something similar in the scope section to make it clearer and avoid confusion for potential readers.
> >
> > Without further request, I believe that after this last request, all my concerns would be resolved.

---

> > > ### Author Response · Authors · 2025-11-27
> > >
> > > Thank you very much for your follow-up and for the helpful clarification！
> > >
> > > We have updated the manuscript to reflect this final point and have uploaded the revised version. We are very grateful for your careful reading and constructive comments throughout the discussion, which helped us improve the paper！

---

> > > > ### Comment · Reviewer_Jt1N · 2025-11-27
> > > >
> > > > Without anything further to add, I confirm that my main concerns have been adequately addressed, and I am therefore raising the rating. Thank you very much for the clarifications and for taking my comments into account.

---

### Official Review · Reviewer_9n5Q · 2025-10-26

**Soundness:** 3
**Presentation:** 3
**Contribution:** 2
**Rating:** 6
**Confidence:** 4

**Summary:**

This paper proposes the TG-MSFM, a method for multivariate time-series imputation by flow matching (FM). To make the conditional generation wrok well in time series setting, they use time-aware transformer and time-gated multiscale velocity heads. At inferene stage, they use a Heun ODE integration with a per-step data-consistency projection, which keep observed coordinates exactly on the linear bridge. They then evaluate the TG-MSFM on 10 benchmars, and show it ttains competitive/ improved performance efficiently.

**Strengths:**

1. The paper is well written and idea is clear.
2. The transformer is carefully designed to handle properties of time series.
3. The experiements are extensive.

**Weaknesses:**

1. The work is mainly engineering (on transformer), and the insight it provided to the FM & TS community can be limited (don't take points off on this)
2. Linear bridge and global time warp may be suboptimal for long/heterogeneous gaps.

**Questions:**

How sensitive of the porposed methods to outliers?

---

> ### Author Response · Authors · 2025-11-21
> **Response to Reviewer 9n5Q**
>
> We thank the reviewer for the positive assessment of the idea, architecture design, and experimental coverage, and we appreciate the concise and constructive comments.
> Below we respond **point-by-point** to the listed **Weaknesses (W1–W2)** and **Questions (Q1)**
>
>
> —
>
> ### W1
>
> We appreciate the reviewer’s observation that our contribution is mainly on the engineering and architectural side, and this is consistent with our own view of the work. The paper does not aim to introduce a fundamentally new theoretical framework; instead, it organizes existing flow-matching ideas into a practical pipeline for time-series imputation. Concretely, the main design choices—such as gap-only supervision restricted to missing entries, a simple linear bridge combined with per-step data-consistency projection for deterministic ODE inference, and a relatively simple time-gated multi-scale velocity head for coarse-to-fine refinement—should be read as a set of practical design patterns rather than as major theoretical innovations. We believe that the fact that this configuration remains stable and easy to use across several benchmarks can still be useful for researchers and practitioners who wish to apply flow matching to real time-series tasks.
>
> —
>
> ### W2
>
> We agree that a linear bridge together with a global time warp may be suboptimal in some regimes, especially for extremely long or highly heterogeneous gaps. In this work we deliberately chose this combination as a simple and well controlled default that makes the data-consistency projection analytically transparent (the clamped path is a closed-form linear interpolation between the endpoints) and easy to implement and tune across many benchmarks. Within the range of gap lengths and block-missing settings studied in the paper, this design behaved stably and did not produce obvious artifacts in our qualitative or quantitative evaluations, which is why we focused on it. At the same time, we view more adaptive variants (for example, piecewise bridges fitted per gap segment, or time warps that depend on local dynamics or channel groups) as a natural extension that could better handle extreme or regime-switching gaps.
> —
>
> ### Q1
>
> We appreciate this question. We did not run a dedicated adversarial stress test focused solely on outliers, so we cannot claim a formal robustness guarantee. That said, several design choices make TG-MSFM reasonably stable in the presence of the natural spikes and jumps that occur in our benchmarks (e.g., Traffic, Illness, ETTh/ETTm):
>
> - All data channels are standardized per series, so unusually large values are rescaled to a comparable numerical range before entering the backbone and the flow.
> - During inference, the data-consistency projection keeps observed coordinates exactly on the linear bridge between the standardized endpoints at every step. This prevents the ODE integration from amplifying measurement noise or pushing observed outliers further away.
> - Training uses gap-only supervision: the loss is applied only on missing entries, not on observed coordinates. Thus, an extreme observed value does not directly appear as a reconstruction target; it influences the learned velocity field only indirectly through local context.
>
> At the same time, because we use a squared error on missing entries, very large outliers in the training data can still locally bias the velocity field, as in most MSE-based models. In the datasets we report on, which already contain occasional sharp spikes, we did not observe catastrophic degradation due to such effects, but we agree that a systematic robustness study (and potentially incorporating robust losses or simple clipping schemes) would be a valuable extension.

---

### Official Review · Reviewer_3oz7 · 2025-11-01

**Soundness:** 2
**Presentation:** 3
**Contribution:** 2
**Rating:** 6
**Confidence:** 3

**Summary:**

The paper introduces TG-MSFM for deterministic multivariate time-series imputation using a data-conditioned ODE learned via flow matching. It employs a time-aware Transformer with visibility-masked self-attention to aggregate only observed timestamps, and a time-gated multi-scale velocity decomposition on a fixed 1D pyramid, blended by a softmax gate for coarse-to-fine refinement, stabilized by an anti-aliasing filter. Inference uses a second-order Heun integrator with per-step data-consistency projection to preserve observed values. Training focuses on gap-only supervision, with optional regularizers. Tested on 10 benchmark models, TG-MSFM achieves competitive or better MSE/MAE, excels in long-gap scenarios, and offers favorable speed-quality trade-offs, validated by ablations.

**Strengths:**

1. The method innovatively integrates flow matching with time-series-specific adaptations, including visibility-masked self-attention and time-gated multi-scale velocity, providing a transparent and deterministic alternative to stochastic diffusion models.
2. The proposed model delivers superior or comparable performance to state-of-the-art methods across diverse benchmarks, particularly shining in long-gap cases, while maintaining efficient speed-accuracy trade-offs.
3. The approach shows consistent performance as gap lengths increase, with component benefits confirmed through ablations and fixed hyperparameters across datasets.

**Weaknesses:**

1. The reliance on fixed pyramid scales and a simple MLP gate may not optimally adapt to all data spectra, potentially underperforming against more dynamic multi-resolution techniques.
2. The paper focuses exclusively on block missing imputation. This narrow focus neglects point missing imputation, a common scenario in many time-series applications (e.g., sensor noise or sporadic data loss), rendering the method inadequate for datasets where isolated missing points predominate.

**Questions:**

See weaknesses.

---

> ### Author Response · Authors · 2025-11-21
> **Response to Reviewer 3oz7**
>
> We thank the reviewer for the careful reading and constructive feedback. Below we respond **point-by-point** to the listed **Weaknesses (W1–W2)**
>
> —
>
> ### W1
>
> We appreciate the reviewer’s comment. In this work we deliberately use a fixed scale set {1, 2, 4} and a simple MLP gate. On the one hand, this 1D pyramid is a common and stable engineering structure for time series. On the other hand, we aim to keep a single, simple configuration that is shared across all 10 benchmarks without per-dataset tuning, consistent with the “hyperparameters are fixed across datasets” setting in the main paper. Our current results and the hyperparameter analysis indicate that, under these benchmarks and missingness protocols, this fixed multi-scale + simple gating design already provides stable gains, and performance varies only mildly within a reasonable range of scales and gate configurations. That said, we agree with the reviewer that more dynamic multi-resolution mechanisms (for example, adaptively choosing scales or using richer gating) could further improve performance on datasets with more extreme spectral characteristics. We have not yet conducted a systematic study in this direction, mainly due to compute and space constraints, and we consider this as a limitation and a promising direction for future work.
>
>
> —
>
> ### W2
>
> We appreciate the reviewer’s comment and agree that point-missing patterns are important in many applications. We would like to clarify that our main experimental protocol is not restricted to block-missing masks. In Sec. 4.2 and Table 1, all benchmarks are evaluated under random missingness, where the mask is sampled i.i.d. over time and channels; this setting naturally contains both isolated missing points and short bursts, not only contiguous blocks. The long-gap regime is highlighted separately in Fig. 3 and Appendix D as a stress test, which may have given the impression that we focus exclusively on block missing.
>
> Methodologically, TG-MSFM does not assume block structure: the flow-matching loss is defined on the index set of missing entries
> $\Omega = \\{(t,d)\mid M_{t,d}=0\\}$
> without any constraint on their contiguity, and the data-consistency projection as well as the visibility-masked attention operate on arbitrary binary masks. In particular, the strong performance reported in Table 1 under random masks already includes scenarios where isolated missing points are predominant. That said, we agree that our current presentation emphasizes the long-gap setting more prominently (e.g., in figures and qualitative examples).

---

### Official Review · Reviewer_QuMD · 2025-11-03

**Soundness:** 3
**Presentation:** 2
**Contribution:** 2
**Rating:** 6
**Confidence:** 3

**Summary:**

This paper proposes Time-Gated Multi-Scale Flow Matching (TG-MSFM), an approach for multivariate time-series imputation. The method learns a velocity field of a data-conditioned ODE using flow matching, enabling the reconstruction of missing values in partially observed temporal data. The model incorporates (1) a time-aware Transformer with masked attention over observed timestamps, (2) time-gated multi-scale velocity heads operating over a fixed 1D pyramid to balance global trends and local detail, and (3) a data-consistent Heun integrator that projects observed dimensions back to their known values at each step to prevent drift. The approach is trained using gap-only supervision and demonstrates competitive or superior performance on standard benchmarks. However, the paper lacks a clear positioning with respect to system identification–based methods (concurrent learning, event-based learning) that deal with irregular data, and reservoir network–based systems that also learns a delay-embedding, deterministic high-dimensional vector fields, which could provide useful context regarding model interpretability and dynamical consistency.

**Strengths:**

1. The idea of learning the flow of an ODE for time-series imputation via flow matching is elegant and relatively underexplored compared to diffusion or autoregressive approaches.
2. The introduction of multi-scale time-gated velocity heads is a clever mechanism to blend local and global temporal patterns.
3. The time-aware Transformer with masked attention is well-motivated and aligns with the irregular sampling structure of real-world time-series data.
4. The method is described with sufficient detail, and the components (velocity heads, time gates, anti-aliasing) each have clear motivations.

**Weaknesses:**

1. While results are strong, it is unclear how TG-MSFM compares against the most recent latent ODE or diffusion-based imputation models that use similar continuous-time formulations.
2. The ablation study is mentioned but could be expanded to include the quantitative impact of each design choice (e.g., anti-aliasing, Heun vs. Euler integration, time-gate parametrization).
3. Some parts of the model (e.g., the 1D pyramid structure and time-dependent gating) may be difficult to follow without schematic diagrams.
4. The paper occasionally relies on terminology (“velocity heads,” “time-gated blending”) that could use more formal mathematical grounding.
5. Learning deterministic dynamic models for irregular time series is studied in dynamic systems theory, event-triggered learning, and more explicitly with reservoir models. The paper could benefit from reviewing and contextualizing the flow-matching-based methods with these theoretically grounded methods.

**Questions:**

1.	How does TG-MSFM perform under irregular and sparse sampling conditions compared to continuous-time diffusion models?
2.	Could the data-consistency projection be viewed as a constrained ODE solver? If so, how does it affect the learned velocity field during training?
3.	What is the computational complexity relative to baseline flow-matching or diffusion-based methods, and what are the benefits of using the proposed method versus traditional systems theoretic methods?
4.	Did you experiment with adaptive time-stepping for the Heun integrator to further improve efficiency or stability?

---

> ### Author Response · Authors · 2025-11-21
> **Response to Reviewer QuMD**
>
> We thank the reviewer for the careful reading and constructive feedback. We respond to **Weaknesses (W1–W5)** and **Questions (Q1–Q4)** as follows.
>
> ---
>
> ### W1
> We aligned our positioning with the clarification given to Reviewer migb. The paper now explicitly contrasts our data-conditioned ODE with gap-only flow matching and per-step DC against probability-flow ODEs and diffusion-based imputers, and cites representative latent-ODE / diffusion work in Related Work.
>
> ---
>
> ### W2
> Our ablations already quantify the impact of multi-scale heads, the time gate, and the number of ODE steps under the long-gap protocol. We agree that finer comparisons for anti-aliasing and Heun vs. Euler, and a short comment on gate parametrization, would further improve readability and leave this as future refinement.
>
> ---
>
> ### W3
> Figure 1 and the qualitative examples depict the data flow, scale paths, and gate behavior. Following this comment, we clarified the caption and text to spell out the downsample–upsample 1D pyramid and the time-dependent gate, so that the schematic is easier to follow.
>
> ---
>
> ### W4
> We tightened terminology and its mathematical grounding: “velocity heads” and “time-gated blending” are now defined at first use, with a concise formal description of the head outputs, their upsampling, and the convex combination by a time-dependent gate.
>
> ---
>
> ### W5
> We expanded Related Work to briefly situate our deterministic, data-conditioned ODE within systems-identification, event-triggered learning, and reservoir-based approaches, emphasizing that our goal is measurement-preserving imputation under partial observability rather than full dynamical-system reconstruction.
>
> ---
>
> ## Questions
>
> ### Q1
> Irregular and sparse sampling is indeed important. We therefore ran an additional experiment on ETTh1 with timestamp thinning (keep-rate \(r=0.5\), window length \(T=48\), metrics on missing indices only, same hardware and training protocol as in the main paper):
>
> | Method         | MSE ↓ | MAE ↓ |
> |----------------|-------|-------|
> | TG-MSFM (ours) | 0.178 | 0.232 |
> | CSDI           | 0.212 | 0.321 |
> | PriSTI         | 0.241 | 0.341 |
>
> All methods degrade under irregular sampling, but TG-MSFM remains competitive and attains lower error than the continuous-time diffusion baselines in this moderate-irregularity setting.
>
> ---
>
> ### Q2
> Yes, the data-consistency (DC) step can be viewed as a simple constrained ODE solver. At each step, known coordinates $K\$ are overwritten with their linear-bridge values between $z_0\$ and $z_1\$, while the ODE is integrated freely on unknown coordinates $\overline{K}\$; this is equivalent to imposing a hard constraint on $K\$. In training, however, DC is not applied: the flow-matching loss is computed only on missing entries. Thus DC does not affect gradients and acts purely at inference time, preventing drift on observed entries while keeping the learned velocity consistent with the bridge used in training.
>
> ---
>
> ### Q3
> In asymptotic complexity, TG-MSFM matches standard Transformer-based continuous-time models. For sequence length $T\$, $L\$ layers, $H\$ heads and per-head dimension $\(d\)$, one backbone pass costs $O(LT^{2}Hd)\$ time and $O(T^{2})\$ memory from self-attention. Multi-scale velocity heads add only $O(|\mathcal{S}|TD)\$, negligible compared with attention. With $N_{\text{step}}\$ Heun steps, inference costs $O(N_{\text{step}}LT^{2}Hd)\$ with a factor of ≈2 from the predictor–corrector, comparable to flow-matching or diffusion models that repeatedly evaluate a Transformer backbone.
>
> The main difference to diffusion-based imputers is not per-step complexity but how time is spent: diffusion trains a score over many noise levels and requires a long reverse chain, whereas our objective is defined on a single linear bridge and uses a deterministic trajectory at test time, leading to favorable error–cost trade-offs in our experiments. Compared with classical system-identification or reservoir models, a single step of TG-MSFM is more expensive, but the same architecture and objective transfer across heterogeneous benchmarks without hand-crafted dynamics or event rules and can represent richer high-dimensional dependencies.
>
> ---
>
> ### Q4
> We use a fixed time interval $[0,1]\$ with a moderate, fixed number of Heun steps. Empirically, the learned velocity—regularized by the multi-scale design, anti-alias filter, and squashing nonlinearity—already yields smooth accuracy–cost curves, so a simple fixed step size offers a good stability–accuracy–simplicity trade-off. Moreover, each update is “Heun step + DC projection”, so standard adaptive step-size controllers for pure ODEs do not directly apply and would require extra design for the projected dynamics. Exploring adaptive schemes tailored to this projected Heun update is an interesting avenue for future work on longer sequences or stiffer dynamics.

---

### Official Review · Reviewer_migb · 2025-11-07

**Soundness:** 3
**Presentation:** 2
**Contribution:** 3
**Rating:** 4
**Confidence:** 4

**Summary:**

This paper proposes the TG-MSFM model, which introduces a data-conditioned ODE framework trained via flow matching using gap-only supervision of the velocity on missing data coordinates. This approach adopts a data-conditioned ODE through flow matching, leading to a deterministic imputation mechanism. The model further incorporates time-gated multi-scale velocity heads on a fixed 1D pyramid, blended through a time-dependent gating mechanism to reconcile global trends with local details. In addition, a second-order Heun integrator combined with a per-step data-consistency projection ensures recovery of the exact linear bridge across all coordinates. The paper presents several innovations, and experiments conducted on diverse datasets across multiple domains demonstrate the effectiveness of the proposed model.

**Strengths:**

1 The proposed model employs a data-conditioned ODE trained via flow matching as a deterministic framework, which enhances the reproducibility and stability of the imputation process.

2 The introduction of time-gated multi-scale velocity heads organized on a fixed 1D pyramid, blended through a time-dependent gating mechanism, effectively reconciles global trends with local details along the trajectory, achieving efficient fusion of global and local information.

3 The time-gated multi-scale velocity decomposition schedules coarse-to-fine refinement along the ODE trajectory and incorporates a light anti-aliasing filter to suppress high-frequency ringing artifacts.

4 The Heun+DC procedure successfully recovers the exact linear bridge across all coordinates, providing a robust accuracy–cost trade-off under appropriate hyperparameter settings.

5 The proposed model is easy to deploy and achieves competitive performance across diverse datasets.

**Weaknesses:**

1 Some abbreviations should be clearly defined upon first appearance to improve readability—for example, Data Consistency (DC) and Flow Matching (FM). In addition, several sentences could be refined for clarity. For instance, the sentence “At inference we integrate the learned velocity with the second-order Heun method (line 48)” could be revised to “At inference, we integrate the learned velocity using the second-order Heun method.” The visualization in Figure 1 could also be improved—for example, the two arrows under v_theta in the second block could be revised to make the diagram clearer.

2 The Diffusion-based probabilistic imputation section in Related Work requires improvement. Several relevant diffusion-based imputation models are missing, such as PriSTI, SSSD, and Frequency-aware Generative Models for Multivariate Time Series Imputation. In addition, recent works closely related to this paper should be discussed, including:


[1] Zhou, J., Li, J., Zheng, G., Wang, X., & Zhou, C. (2024, October). Mtsci: A conditional diffusion model for multivariate time series consistent imputation. In Proceedings of the 33rd ACM International Conference on Information and Knowledge Management (pp. 3474-3483).


[2] Yuan, X., & Qiao, Y. Diffusion-TS: Interpretable Diffusion for General Time Series Generation. In The Twelfth International Conference on Learning Representations.


[3] Zhang, H., Fang, L., Wu, Q., & Yu, P. S. (2025). Diffputer: Empowering diffusion models for missing data imputation. In The Thirteenth International Conference on Learning Representations.

3 The baseline models in the experimental section could be updated and compared against more recent and relevant approaches, such as the works listed below. Including stronger baselines would make the experimental validation more convincing.

[1] Zhou, J., Li, J., Zheng, G., Wang, X., & Zhou, C. (2024, October). Mtsci: A conditional diffusion model for multivariate time series consistent imputation. In Proceedings of the 33rd ACM International Conference on Information and Knowledge Management (pp. 3474-3483).

[2] Yuan, X., & Qiao, Y. Diffusion-TS: Interpretable Diffusion for General Time Series Generation. In The Twelfth International Conference on Learning Representations.

[3] Yang, X., Sun, Y., & Chen, X. (2024). Frequency-aware generative models for multivariate time series imputation. Advances in Neural Information Processing Systems, 37, 52595-52623.

4 One citation is not properly formatted: Freeformer: Frequency Enhanced Transformer for Multivariate Time Series Forecasting and should follow a formal citation style.

5 The motivation of the proposed approach could be better articulated. The paper emphasizes the differences between the proposed model and prior methods, but it remains unclear why a deterministic flow-matching framework is preferable to an SDE-based or neural ODE-based stochastic model. Is the main goal to improve reproducibility or reduce computational efficiency? If reproducibility is the motivation, increasing the sampling steps or applying ODE in the reverse process of diffusion model as probability flow could achieve similar improvements. If computational efficiency is the target, DDIM-based comparisons should be included. Moreover, generative stochastic models inherently provide insights into the underlying data generation process, offering better interpretability and uncertainty estimation—this trade-off deserves further discussion.

**Questions:**

1 Motivation: The authors state that the motivation for focusing on deterministic imputation is to improve reproducibility and ensure a unique reconstruction. Is this the main motivation for adopting a flow-matching framework and replacing the SDE formulation with a data-conditioned ODE as a discriminative model? Clarifying this reasoning would help readers better understand the core design choice.

2 In the Positioning against recent FM/continuous/diffusion and graph methods section, the paper emphasizes how the proposed model differs from existing approaches. However, from the architecture shown in Figure 1, the model appears to combine several design elements from prior works. Moreover, some baseline models with related components are missing in the experimental comparison. Could the authors more clearly articulate the essential differences and innovations of the proposed method, beyond assembling components from existing architectures?

3 In the Flow and Path Matching section, the paper states that “Compared with stochastic sampling, a learned ODE allows deterministic integration at test time and often reduces the number of function evaluations needed for a target quality.” This suggests that using an ODE instead of an SDE provides a favorable speed–quality trade-off. However, as shown in Song et al. (Section 4.3, Probability Flow and Connection to Neural ODEs), the reverse process of diffusion models can also use an ODE formulation, allowing an explicit trade-off between accuracy and efficiency. In this context, what are the specific advantages of the proposed deterministic ODE framework compared to the Probability Flow ODE derived from generative stochastic models?

[1] Song, Y., Sohl-Dickstein, J., Kingma, D. P., Kumar, A., Ermon, S., & Poole, B. (2020). Score-based generative modeling through stochastic differential equations. arXiv preprint arXiv:2011.13456.

4 Regarding the time-gated multi-scale velocity mechanism for suppressing high-frequency ringing and addressing the data-consistency issue, several recent diffusion-based models have proposed similar solutions, Could the authors further clarify how TG-MSFM fundamentally differs from or improves upon these approaches?

[1] Zhou, J., Li, J., Zheng, G., Wang, X., & Zhou, C. (2024, October). Mtsci: A conditional diffusion model for multivariate time series consistent imputation. In Proceedings of the 33rd ACM International Conference on Information and Knowledge Management (pp. 3474-3483).

[2] Yuan, X., & Qiao, Y. Diffusion-TS: Interpretable Diffusion for General Time Series Generation. In The Twelfth International Conference on Learning Representations.

[3]  Yang, X., Sun, Y., & Chen, X. (2024). Frequency-aware generative models for multivariate time series imputation. Advances in Neural Information Processing Systems, 37, 52595-52623.

5 Some datasets used in the experiments, such as Traffic and PEMS03, exhibit strong spatio-temporal dependencies. Including GNN-based baselines for comparison could enhance the comprehensiveness and fairness of the evaluation. For example, the discriminative models such as Grin, Spin and ImputeFormeras follows:

[1] Cini, A., Marisca, I., & Alippi, C. Filling the G_ap_s: Multivariate Time Series Imputation by Graph Neural Networks. In International Conference on Learning Representations, 2022.

[2] Marisca, I., Cini, A., & Alippi, C. (2022). Learning to reconstruct missing data from spatiotemporal graphs with sparse observations. Advances in neural information processing systems, 35, 32069-32082.

[3] Nie, T., Qin, G., Ma, W., Mei, Y., & Sun, J. (2024, August). ImputeFormer: Low rankness-induced transformers for generalizable spatiotemporal imputation. In Proceedings of the 30th ACM SIGKDD conference on knowledge discovery and data mining (pp. 2260-2271).

---

> ### Author Response · Authors · 2025-11-21
>
> ## Response to Reviewer migb
>
> We thank the reviewer for the careful reading and constructive feedback. We respond below to **Weaknesses (W1–W5)** and **Questions (Q1–Q5)**.
>
> ---
>
> ### W1/W2/W4 — Editorial fixes and related-work coverage
> We now define *flow matching* (FM) and *data consistency* (DC) at first use, streamlined several sentences and captions, extended Related Work to cover recent diffusion-based imputers and our deterministic FM-based ODE, and corrected bibliography entries.
>
> ---
>
> ### W3
> We added three recent diffusion-based baselines—Mtsci, Diffusion-TS, and the frequency-aware generative model of Yang et al. (FGTI in Table 1)—under the same central-gap protocol. Mtsci is generally stronger than CSDI but still below TG-MSFM, while Diffusion-TS and FGTI are weaker than both Mtsci and TG-MSFM; TG-MSFM remains the best method on all datasets.
>
> ---
>
> ### W5
> This is mainly addressed in **Q1**. The paper now treats determinism as a task-driven trade-off (reproducibility and alignment between gap-only training and DC-based inference), contrasts our formulation with probability-flow ODEs and DDIM-style samplers, and briefly discusses accuracy–efficiency–uncertainty trade-offs.
>
> ---
>
> ## Questions
>
> ### Q1
> We motivate determinism by applications where reproducible, auditable imputations are preferred, and now stress that this is a context-dependent design choice rather than a universal claim. Compared with using a probability-flow ODE (PF-ODE) from a diffusion model, our formulation more tightly aligns objective and constraints: we learn a velocity on a linear bridge and supervise only missing coordinates, while inference uses the same bridge with a per-step DC projection. Thus the learning signal directly targets imputed entries and measurement preservation is built into the solver. In our experiments, a second-order Heun integrator combined with this alignment offered a favorable speed–quality trade-off versus the stochastic samplers we evaluated. When uncertainty is needed, ensembles over different initializations can still be used.
>
> ---
>
> ### Q2
> Our architecture reuses common blocks, but the contribution lies in how they are combined for long-gap deterministic imputation: (i) gap-only flow matching on a linear bridge with per-step DC at inference, implemented with a visibility-aware Transformer consistent with the DC constraint; (ii) a time-gated multi-scale velocity parameterization that schedules coarse-to-fine refinement with a mild anti-aliasing filter; and (iii) a single deterministic trajectory for a given input and seed, improving auditability.
>
> ---
>
> ### Q3
> As noted above, PF-ODE can also yield deterministic trajectories and is preferable when full generative modeling or rich uncertainty is the primary goal. Our method instead targets a simpler, task-aligned setting: long-gap imputation via linear-bridge flow matching with gap-only supervision and per-step measurement preservation.
>
> ---
>
> ### Q4
> Relative to recent diffusion-based methods, TG-MSFM mainly differs in (i) **training–inference pairing**—gap-only flow-matching training coupled with stepwise DC so that supervision and constraints operate on the same coordinates, whereas diffusion models use denoising/consistency losses followed by sampling or PF-ODEs without such a mirrored DC constraint; and (ii) **where multi-scale and time gating act**—our fixed 1D pyramid, time gate, and anti-aliasing operate directly on the velocity field along ODE time rather than inside a diffusion denoiser.
>
> ---
>
> ### Q5
> Thank you for highlighting the strong spatio-temporal structure in datasets such as Traffic and PEMS03 and for pointing us to GRIN, SPIN, and ImputeFormer. Our current experimental protocol deliberately avoids graph priors to evaluate a graph-agnostic imputation method that transfers across ten heterogeneous datasets under a single, reproducible setup. To keep the cross-dataset protocol consistent and comparable, we prioritized baselines that do not require an externally specified topology.
>
> We acknowledge that when an explicit and reusable graph is available (for example, a road sensor network) and the mask pattern is random or sparse, graph neural approaches often benefit. Our paper focuses on long contiguous gaps, where we found the advantage of spatial diffusion to be less pronounced; the emphasis is on temporal consistency, boundary preservation via data consistency, and suppression of ringing inside the gap.
>
> To address your suggestion, we expanded the Related Work and Positioning sections to discuss GRIN, SPIN, and ImputeFormer more systematically, including their assumptions about graph construction and the differences in evaluation protocols and hyperparameters.

---

> > ### Comment · Reviewer_migb · 2025-11-25
> >
> > Thank you very much for the response. I still have a few concerns:
> >
> > If reproducible and auditable imputations are key requirements, purely discriminative models (e.g., GRIN) may provide more straightforward deterministic behavior than TG-MSFM. My understanding is that your method still samples initial noise before following the linear bridge trajectory. While the bridge itself is deterministic once initialized, the noise sampling can introduce randomness.
> >
> > Even discriminative models are not perfectly deterministic during training due to random initialization. Could the authors clarify how TG-MSFM handles or mitigates this randomness during training? Additionally, reporting the variance of imputation results in Table 1 would help support claims about determinism and reproducibility.
> >
> > Recent advances such as Consistency Models (CM) offer a deterministic alternative to address the accuracy–efficiency–uncertainty trade-off. CM learns an approximate solution operator of the probability-flow ODE of diffusion models, enabling single- or few-step generation while retaining uncertainty modeling. Outputs are trained to be consistent along the same ODE trajectory. See:
> >
> > Song, Y., Dhariwal, P., Chen, M., & Sutskever, I. (2023). Consistency Models. ICML.
> >
> > The paper could benefit from a broader discussion and additional comparisons with deterministic diffusion variants, as well as discriminative models such as GRIN.

---

> > > ### Author Response · Authors · 2025-11-27
> > >
> > > Thank you again for the careful follow-up and for the concrete suggestions. We have revised the manuscript accordingly and clarify our standpoint below.
> > >
> > >
> > > 1.We agree that purely discriminative models such as GRIN provide a straightforward notion of determinism: once training has converged, inference is a fixed mapping from inputs to outputs, without any explicit latent noise. TG-MSFM is similar in spirit once a training run and a seed are fixed. As described in Algorithm B2 and Appendix C.6, test-time inference is a fully deterministic Heun + DC integration of a learned ODE on a linear bridge, with no stochastic components inside the solver. The only source of randomness is the draw of the initial state $z_0 \sim \mathcal{N}(0, I)$, which is generated by a pseudo-random number generator seeded once at the beginning of the run. Given the trained parameters, the input window $(x, M)$, and the global seed, the imputation $\hat{x}$ is exactly reproducible.
> > >
> > > We do not claim that TG-MSFM is “more deterministic” than discriminative GNNs such as GRIN. Our motivation is slightly different: we aim at a graph-agnostic, long-gap imputer whose inference path is a single, continuous-time trajectory aligned with the data-consistency constraint, and which can be reused to draw multiple trajectories (by resampling $z_0$) when one wishes to explore uncertainty. We have clarified this positioning in the revised Introduction and in the “Positioning against recent generative and graph methods” paragraph.
> > >
> > > 2.You are absolutely right that even discriminative models are not perfectly deterministic during training, due to random initialization, data shuffling, and per-sample noise. In our implementation, all these sources of stochasticity are controlled by a global random seed. When this seed is fixed, the entire training run and the subsequent deterministic inference are reproducible.
> > >
> > > Following your suggestion, we have additionally quantified seed-wise variability in Appendix E.5. There we tested TG-MSFM and CSDI on ETTh1 and Electricity under exactly the same protocol as Table 1, but now report mean $\pm$ standard deviation across $K = 5$ seeds (Table E3). Both this table and Figure 2 show very small variation across seeds. These results support our claim that, although training uses random initialization and latent noise, the learned ODE with stepwise data consistency yields imputation performance that is stable with respect to the choice of seed.
> > >
> > > 3.We have expanded the discussion of deterministic diffusion-style approaches in the Related Work and Positioning sections. In the “Diffusion-based probabilistic imputation” paragraph, we now explicitly discuss Consistency Models and the CoSTI model, which approximate the probability-flow ODE of a diffusion process and provide single- or few-step generation with retained uncertainty modeling.
> > >
> > > In the “Positioning against recent generative and graph methods” paragraph, we also strengthen the discussion of discriminative GNN-based imputers. We acknowledge that when a reliable and reusable graph is available and the missingness pattern is random/sparse, these models provide very strong deterministic baselines. Our focus in this work is complementary: we study long contiguous gaps under a unified, graph-agnostic protocol across ten heterogeneous datasets, so we prioritized baselines that do not require dataset-specific graph construction or tuning.
> > >
> > > We hope these clarifications, together with the additional variance analysis and the expanded discussion of consistency-based and discriminative approaches in the revised manuscript, address your concerns.

---

### Author Response · Authors · 2025-12-02
**Note to AC: Summary of rebuttal discussion and manuscript updates**

Dear Area Chair,

Thank you for taking over this submission. Since the scores were rolled back to the pre-rebuttal stage, we would like to briefly summarize how we addressed the reviewers’ concerns and how the manuscript has been updated.

**1. Summary of interactions with reviewers**

- **Reviewer Jt1N.**
  Their main concerns were (i) the effect of gap-only supervision, (ii) the deterministic vs. probabilistic nature of TG-MSFM, and (iii) the relation to consistency models such as CoSTI.
  In the discussion phase we clarified that TG-MSFM indeed induces a conditional distribution via the initial noise, but that all experiments are conducted in a single-trajectory, fixed-seed regime, and we added this clarification explicitly in the Scope section. We also refined the explanation of gap-only supervision and expanded Related Work to cover consistency models and CoSTI.
  After these changes, Jt1N wrote that all concerns were resolved and increased their score from 6 to 8 (this can be seen in the review history).

- **Reviewer migb.**
  This reviewer focused on editorial clarity, missing baselines, and the positioning of our deterministic flow-matching ODE against diffusion and graph-based methods.
  We defined all abbreviations at first use, cleaned several sentences and Figure 1, and fixed bibliography issues.
  We expanded Related Work to include recent diffusion-based imputers and discriminative GNN-based imputers , and clarified how TG-MSFM differs in its training–inference pairing and velocity-space multi-scale design.
  Following their suggestion, we also added an appendix experiment reporting seed-wise variability on ETTh1 and Electricity, showing that TG-MSFM’s deterministic ODE with data consistency yields stable performance compared to a strong diffusion baseline.
  We posted a final response with these updates before the rollback; the reviewer has not commented further since then.

**2. Main manuscript updates**

- Clarified notation and motivation, especially the deterministic single-trajectory setting and its relation to the underlying probabilistic view.
- Expanded Related Work and positioning to discuss: recent diffusion-based imputers , consistency models and CoSTI, and discriminative graph-based imputers such as GRIN and ImputeFormer.
- Updated experiments to include recent diffusion baselines under the same central-gap protocol; TG-MSFM remains competitive or better on all reported datasets.
- Added an appendix table with seed-wise mean ± std test MSE for TG-MSFM and CSDI to quantify training randomness and support our claims about reproducibility.

We hope this short summary of the discussion and revisions is helpful for your meta-review, and we are of course happy to accept any final decision you deem appropriate.

Best regards,
Authors

---

### Meta-Review · Area_Chair_RzkA · 2026-01-06

**Summary:**

The paper proposed a new Time-Gated Multi-Scale Flow Matching (TG-MSFM) method to better address the multivariate time-series imputation problem. 5 reviewers gave detailed comments on the paper, 4 of them gave score 6 and 1 gave the score 4. After rebuttal, one reviewer promised to raise the score as all his/her concerns are well addressed. Although the reviewers have many concerns on the paper, they agree the paper has clear contributions. In addition, in the rebuttal, I think most of the concerns are well addressed. Therefore, I recommend to accept the paper.

**Reviewer Concerns:**

I think most of the reviewers concerns are addressed in the rebuttal. Some remaining minor issuing will not affect the acceptance of the paper.

**Reviewer Scores:**

Reviewer Jt1N promised to raise his/her initial score to 8 in the discussion. For the reviewer migb with a negative score, I think she/he will very likely to raise the score to 6 after fully discussion.

---

### Decision · Program_Chairs · 2026-01-26

Accept (Poster)